# Gap junctions amplify spatial variations in cell volume in proliferating tumor spheroids

Eoin McEvoy [1,2], Yu Long Han[3], Ming Guo [3] & Vivek B. Shenoy [1,2✉]

Sustained proliferation is a significant driver of cancer progression. Cell-cycle advancement is coupled with cell size, but it remains unclear how multiple cells interact to control their volume in 3D clusters. In this study, we propose a mechano-osmotic model to investigate the evolution of volume dynamics within multicellular systems. Volume control depends on an interplay between multiple cellular constituents, including gap junctions, mechanosensitive ion channels, energy-consuming ion pumps, and the actomyosin cortex, that coordinate to manipulate cellular osmolarity. In connected cells, we show that mechanical loading leads to the emergence of osmotic pressure gradients between cells with consequent increases in cellular ion concentrations driving swelling. We identify how gap junctions can amplify spatial variations in cell volume within multicellular spheroids and, further, describe how the process depends on proliferation-induced solid stress. Our model may provide new insight into the role of gap junctions in breast cancer progression.

[1] Center for Engineering Mechanobiology, University of Pennsylvania, Philadelphia, PA, USA. [2] Department of Materials Science and Engineering, University of Pennsylvania, Philadelphia, PA, USA. [3] Department of Mechanical Engineering, MIT, Cambridge, MA, USA. ✉email: vshenoy@seas.upenn.edu

Cell volume is typically tightly regulated to sustain normal function and survival[1]. In single cells, volume control involves an interplay between ion channels on the membrane that permit passive exchange of solutes between the cytosol and extracellular fluid and active ion pumps that move solutes against a concentration gradient[2,3]. As water movement across the cell membrane is largely driven by osmotic pressure, precise control of the cytosolic ion concentration can increase or decrease cell volume[4]. Impairment of ion regulation has severe consequences and is indicated in many disease states; for example, sickle cell dehydration is associated with a pathological loss of erythrocyte ions[5], and potassium channels play a role in cells resisting apoptosis during cancer development[6]. In multicellular systems, cells adhere to one another via cadherin- and catenin-mediated complexes[7]. Alongside these adhesions, connexin structures assemble to form gap junctions (GJs) that permit exchange of ions and fluid between cells[8]. Although these junctions are particularly appreciated to be of importance during development[9] and in cardiac conductance[10], they are present in the majority of mammalian cell types. However, despite clear intuition that water and ion flow across GJs should confound cytosolic osmolarity and influence cellular shrinkage and swelling, their role in volume dynamics has not been well studied.

In exploring cell behavior, multicellular spheroids have emerged as an increasingly promising experimental model that aim to bridge the gap between in vitro and in vivo conditions[11]. Recently, we seeded mammary epithelial cells in hydrogel to investigate how individual cell volumes vary spatially in a proliferating cluster[12]. We identified that peripheral cells became more swollen as the cluster grew and cells at the core reduced in size. Blocking GJs normalized volume distributions, indicating that they play an important role in mediating differential cell swelling. The development and progression of cancer is driven by a myriad of factors, including matrix density[13,14], cell adhesion[15,16], and interactions between different cell types[17,18], that collectively influence cell proliferation, apoptosis, and matrix invasion. Furthermore, proliferation also depends on cellular size[19,20] and stress[21,22]. However, in 3D clusters it remains unclear how individual cells coordinate to regulate their volumes.

Motivated by early work on water movement across lipid membranes[23], a number of analytical models have sought to address how cells regulate their volume via ion exchange with their microenvironment[24–28]. Jiang and Sun[29] considered the critical role of cell mechanics and ion channel mechano-sensitivity, highlighting how cells can maintain their volume in response to osmotic shock. Beyond, similar models have been proposed to understand how fluid or ion transport governs the swelling or shrinkage of adhered single cells[30,31] and lumen growth[32,33]. To our knowledge, however, the role of ion exchange across GJs within multicellular systems has not yet been investigated. We hypothesize that GJs play a key role in the regulation of volume in connected cells, amplifying spatial variations in cell volume by mediating solute flow in response to mechanical loading. We therefore propose a mechano-osmotic model for the analysis of fluid and ion transport between connected cells and their environment. Initially considering a simple two-cell system, we demonstrate that when a cell experiences increased solid stress loading, evolving osmotic pressure gradients drive swelling of its connected neighbor. We then expand our framework to explore how GJs amplify spatial variations in cell volume across a multicellular spheroid, highlighting an interplay between non-uniform proliferation-driven stress, cell mechanics, and transmembrane ion flow.

## Results

### A mechano-osmotic model for cellular volume control that integrates mechanical force balance with fluid and ion fluxes.

To approach the problem of multicellular volume regulation, we initially consider two cells (Fig. 1) held together via cadherin- and catenin-mediated complexes. As these complexes stabilize on the membrane, connexin structures also assemble and couple with identical units on the neighboring cell to form GJs. These channels connect the cytoplasm of both cells, permitting passive transport of fluid, ions, and small molecules[34]. GJs typically remain open during their lifecycle, though may close in response to high $Ca^{2+}$ concentrations or low pH which serves to protect the cell from dying neighbors[35]. Importantly, movement of fluid and ions not only depends on GJ-mediated transport, but also on passive channels and active ion pumps within the cell membrane that permit transfer to and from the extracellular environment.

**GJ-mediated ion and fluid transport between cells.** To understand how water and ions move between the two cells, we first consider the role of hydrostatic and osmotic pressure, denoted by $P$ and $\Pi$, respectively. A detailed derivation for our thermodynamic framework is provided in Supplementary Notes 1–3, with the key governing equations briefly discussed here. GJs have a diameter in the range of 1.5−2 nm, and can be approximated as fully non-selective to water molecules (diameter $\approx 0.275$ nm) and ions (diameters $\approx 0.1 - 0.2$ nm). Movement of water between cells is driven by a difference in hydrostatic pressure (see Supplementary Note 2 for motivation) such that the volume flux across GJs (from cell $i+1$ to cell $i$) is described by $J_{v,g,i} = -L_{p,g}(P_i - P_{i+1})$, where $L_{p,g}$ is a constant that relates to the water permeability of GPs. We can therefore assume changes in cell volume $V_{c,i}$ are given by:

$$\frac{dV_{c,i}}{dt} = -A_g L_{p,g}(P_i - P_{i+1}), \tag{1}$$

where $A_g$ is the surface area of the membrane connected to the neighboring cell. GJs also permit a flow of ions between cells, as driven by diffusive and advective flow (see Supplementary Note 2). Assuming dilute conditions, the cell's internal osmotic pressure relates to the number of ions $N_i$ in the cytosol via Van't Hoffs equation $\Pi_i = N_i RT/V_{c,i}$, where $R$ is the gas constant and $T$ is the absolute temperature. The advective/diffusive behavior may then be characterized by an ion flow given by $\dot{n}_{g,i} = -c_s^* L_{p,g}(P_i - P_{i+1}) - \omega_g(\Pi_i - \Pi_{i+1})$, where $\omega_g$ is rate constant and $c_s^*$ is the mean solute concentration across the two connected cells. Under these conditions, the rate of change in the total number of ions in a cell can be determined:

$$\frac{dN_i}{dt} = -A_g\left(c_s^* L_{p,g}(P_i - P_{i+1}) + \omega_g(\Pi_i - \Pi_{i+1})\right). \tag{2}$$

**Mechanics of the cell cortex.** As water enters a cell, the increase in fluid volume stretches the cell membrane. The mechanical tension in the membrane is complex, controlled by membrane–cytoskeleton adhesion, cortical stiffness, and active myosin contractility[36,37]. We treat the membrane and cortex as a single mechanical structure[29], neglecting the possibility of cortical detachment and blebbing. The constitutive law of the cortical structure can be written as $\sigma_i = \sigma_{p,i} + \sigma_{a,i}$, where $\sigma_{a,i}$ is the active stress associated with myosin contractility and $\sigma_{p,i}$ is the passive stress predominantly associated with deformation of the actin network (as the actin cortex is much stiffer than the plasma membrane[36,38]). With the assumption that the passive stress increases linearly with stretch, it can be expressed as $\sigma_{p,i} = K(A_{c,i}/A_{c,i}^0 - 1)/2$, where $K$ is the effective stiffness, $A_{c,i}$ the surface area of the cell, and $A_{c,i}^0$ a reference surface area. In addition to internal fluid pressure, the membrane also experiences loading from a spatially uniform external fluid pressure $P^{ext}$. Mechanical force balance for a spherical cell with radius $r_{c,i}$

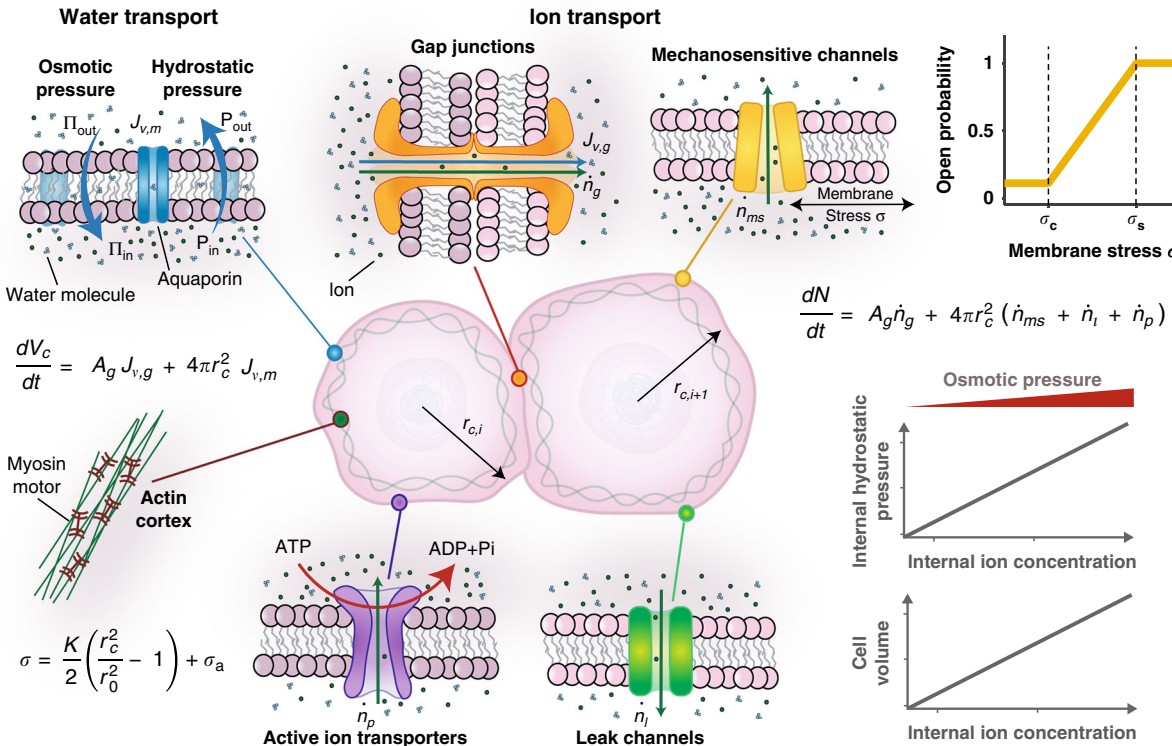

**Fig. 1 Schematic of cellular model.** Water moves across the semi-permeable cell membrane as driven by hydrostatic or osmotic pressure gradients. Ions can diffuse between cells through gap junctions and across the membrane through mechanosensitive and leak channels. Active pumps move ions into the cell against a concentration gradient.

dictates that the cortical stress can be related to the pressure difference across the membrane $\Delta P_i = P_i - P^{\text{ext}}$. Therefore, the cortical stress may also be written as $\sigma_i = \Delta P_i r_{c,i}/2h_i$, where $h_i$ is the cortical thickness. Further, within a multicellular organoid, proliferation of cells generates compressive solid stresses $\sigma_{g,i}$ that act on neighboring cells[39]. Deformation of fibrous matrix surrounding the cell cluster compounds the stress, as stretched fibers squeeze on the cluster[40]. Thus, we obtain the following expanded expression for the membrane/cortical stress:

$$\sigma_i = \frac{K}{2}\left(\frac{r_{c,i}^2}{r_0^2} - 1\right) + \sigma_{a,i} = \frac{(\Delta P_i - \sigma_{g,i})r_{c,i}}{2h_i}, \quad (3)$$

where $r_0$ is the reference cell radius. Additional details on our treatment of dissipative behavior and model linearization are discussed in Supplementary Note 4.

**Fluid and ion exchange with the extracellular environment.** In addition to diffusion across GJs, water molecules can move through the semi-permeable cell membrane, enhanced by the presence of aquaporins[41]. As these pathways do not permit the diffusion of ions, we can consider them to be fully selective. We assume that the ion concentration in the external media is uniform, such that the external osmotic pressure at any point is given by $\Pi^{\text{ext}}$ (non-uniform external ion concentrations are explored in Supplementary Note 9). Therefore, the membrane water flux can be expressed as $J_{\text{v,m},i} = -L_{\text{p,m}}(\Delta P_i - \Delta\Pi_i)$, where $\Delta P_i = P_i - P^{\text{ext}}$ and $\Delta\Pi_i = \Pi_i - \Pi^{\text{ext}}$, and $L_{\text{p,m}}$ is the permeability coefficient associated with solvent flow through the membrane. Evidently, this flux depends on the difference in osmotic and hydrostatic pressure between the cell and the extracellular environment. We can then extend Eq. 1 to consider this additional water flux such that $dV_{c,i}/dt = A_g J_{\text{v,g},i} + A_i J_{\text{v,m},i}$, where $A_i = 4\pi r_{c,i}^2$ is the cell surface area. Note that with our assumption of uniform external

hydrostatic and osmotic pressures (e.g. $P_i^{\text{ext}} = P_{i+1}^{\text{ext}} = P^{\text{ext}}$), we can also state the GJ water flux as a function of pressure differences, such that $J_{\text{v,g},i} = -L_{\text{p,g}}(\Delta P_i - \Delta P_{i-1})$. Assuming the cells can be approximated to retain a spherical shape with radius $r_{c,i}$, we achieve the following expanded form for cellular volume change:

$$\frac{dV_{c,i}}{dt} = -A_g L_{\text{p,g}}(\Delta P_i - \Delta P_{i+1}) - A_i L_{\text{p,m}}(\Delta P_i - \Delta\Pi_i). \quad (4)$$

The cytosolic ion concentration also depends on exchange with the extracellular environment through selective ion channels. As these channels do not facilitate solvent flow, the associated fluxes assume the general form $\dot{n}_i = \omega_j \Delta\Pi_i$. Mechanosensitive (MS) channels are proteins in the cell membrane that open under a tensile membrane stress[42] to allow flow of ions from regions where the concentration is high to regions where it is low. As an example, Piezo1 opens under tension to allow a calcium influx, thereby activating $Ca^{2+}$-gated $K^+$ channels to relieve osmotic pressure in swollen cells[43]. The probability of channel opening has been reported to follow a Boltzmann function[44], and consistent with Jiang and Sun[29], we adopt a piecewise linear expression (Fig. 1, yellow curve) to describe the ion flux associated with MS channel permeability $\dot{n}_{\text{ms},i} = -\omega_{\text{ms}}(\sigma_i)\Delta\Pi_i$, such that

$$\omega_{\text{ms}}(\sigma_i) = \begin{cases} 0 & \text{if } \sigma_i \le \sigma_c \\ \beta(\sigma_i - \sigma_c) & \text{if } \sigma_c < \sigma_i < \sigma_s, \\ \beta(\sigma_s - \sigma_c) & \text{if } \sigma_i \ge \sigma_s \end{cases} \quad (5)$$

where $\sigma_i$ is the computed cortical stress, $\sigma_c$ is the threshold stress, below which $\dot{n}_{\text{ms},i} = 0$, $\sigma_s$ is the saturating stress, above which the channels are fully open, and $\beta$ is a rate constant. In addition to these force sensitive channels, there are a number of leak channels (which are always operative) on the membrane[2] for which we

consider a further transmembrane ion flux $\dot{n}_{l,i} = -\omega_l \Delta \Pi_i$, where $\omega_l$ is the associated permeability coefficient.

While the channels described thus far permit passive ion diffusion, there are additional membrane proteins present that actively transport ions against the concentration gradient. These ion pumps require an energy input, such as from ATP hydrolysis, to overcome the energetic barrier associated with moving ions against the concentration gradient. Following Jiang and Sun[29], the free energy change associated with pumping action can be expressed as $\Delta G = RT \log(c_i/c_{ext}) - \Delta G_a$, where $\Delta G_a$ is an energy input is associated with hydrolysis of ATP. The ion flux associated with active pumping can then be written as $\dot{n}_{p,i} = \gamma' \Delta G$, where $\gamma'$ is a permeation constant. Maintaining our dilute assumption, $\Delta G$ can be linearized as $\Delta G = RT(\Pi_i - \Pi^{ext})/\Pi^{ext} - \Delta G_a$. We can therefore identify a critical osmotic pressure difference $\Delta \Pi_c$, determined when $\Delta G = 0$, such that $\Delta \Pi_c = \Pi^{ext} \Delta G_a / RT$ (noting that when $\Delta G > 0$ active pumping is no longer energetically favorable and the pumping direction will reverse[45]). Thus, the ion flux generated by active pumping can be expressed as $\dot{n}_{p,i} = \gamma(\Delta \Pi_c - \Delta \Pi_i)$, where $\gamma$ is a rate constant. In this framework as we only consider a single ion species, we neglect the influence of electroneutrality and membrane potential. However, a detailed analysis of these additional mechanisms is provided in Supplementary Note 8 where we identify that our simplified approach predicts similar trends to a full electro-osmotic 'pump-leak' framework (Supplementary Figs. 5 and 6). Taking pumps and channels into consideration, we can then extend Eq. 2 for a more detailed description of the number of ions within the cell whereby $dN_i/dt = A_g \dot{n}_{g,i} + A_i(\dot{n}_{ms,i} + \dot{n}_{l,i} + \dot{n}_{p,i})$. The mean ion concentration between connected cells, $c_s^*$, can be expressed in terms of osmotic pressure as $c_s^* = (c_i + c_{i+1})/2 = (\Pi_i + \Pi_{i+1})/(2RT)$. Based on our established terminology whereby $\Delta \Pi_i = \Pi_i - \Pi^{ext}$, this can be rephrased such that $c_s^* = (\Delta \Pi_i + \Delta \Pi_{i+1} + 2\Pi^{ext})/(2\,RT)$. Assuming that $\Delta \Pi_i/\Pi^{ext} \ll 1$, we can therefore approximate

$c_s^* \approx \Pi^{ext}/RT$ and the number of cellular ions can be characterized by

$$\frac{dN_i}{dt} = -A_g \left( \frac{\Pi^{ext}}{RT} L_{p,g} (\Delta P_i - \Delta P_{i+1}) + \omega_g (\Delta \Pi_i - \Delta \Pi_{i+1}) \right) \\ - A_i((\omega_{ms}(\sigma_i) + \omega_l + \gamma)\Delta \Pi_i - \gamma \Delta \Pi_c). \quad (6)$$

Material parameters for all simulations are summarized in Table S1.

## GJs amplify differential swelling associated with proliferation-induced solid stresses in neighboring cells.

Typically, cell clusters are seeded in a confining matrix, and as the cluster grows it displaces and deforms the elastic matrix, opposing growth and generating solid stresses within the cluster. When cells proliferate within the growing cluster, they push against and apply compressive stresses on their neighbors[39], and with the addition of cell adhesion and local jamming this leads to different levels of stress developing spatially. To understand the influence of such solid growth stress on cellular shrinkage and swelling, we first consider the interactions between two connected cells (Fig. 2a). Without loss of generality, we assume there to be a compressive stress $\sigma_{g,1} = \sigma_{g,0} + \delta \sigma_{g,1}$ acting uniformly on the surface of one cell and for its neighbor to be acted upon by a stress $\sigma_{g,2} = \sigma_{g,0}$. For the purpose of illustration, we choose $\sigma_{g,0}$ to equal zero and for $\delta \sigma_{g,1}$ to increase over time to a maximum of 150 Pa (Supplementary Fig. 1a). Varying these parameters will lead to similar trends albeit different magnitudes.

When GJs are active (control case), our model predicts that the loaded cell shrinks and its neighbor swells (Fig. 2b). Initially, in the absence of loading, we find that cells control their volume by regulating their ion concentration through the activities of ion pumps and channels (Fig. 2b and Supplementary Fig. 1). In this unloaded state, MS channels are permeable due to tension in the cell membrane, permitting a constant loss of ions to the external media (Supplementary Fig. 1d). However, active ion pumping ensures there is a continuous ion influx (Supplementary Fig. 1f)

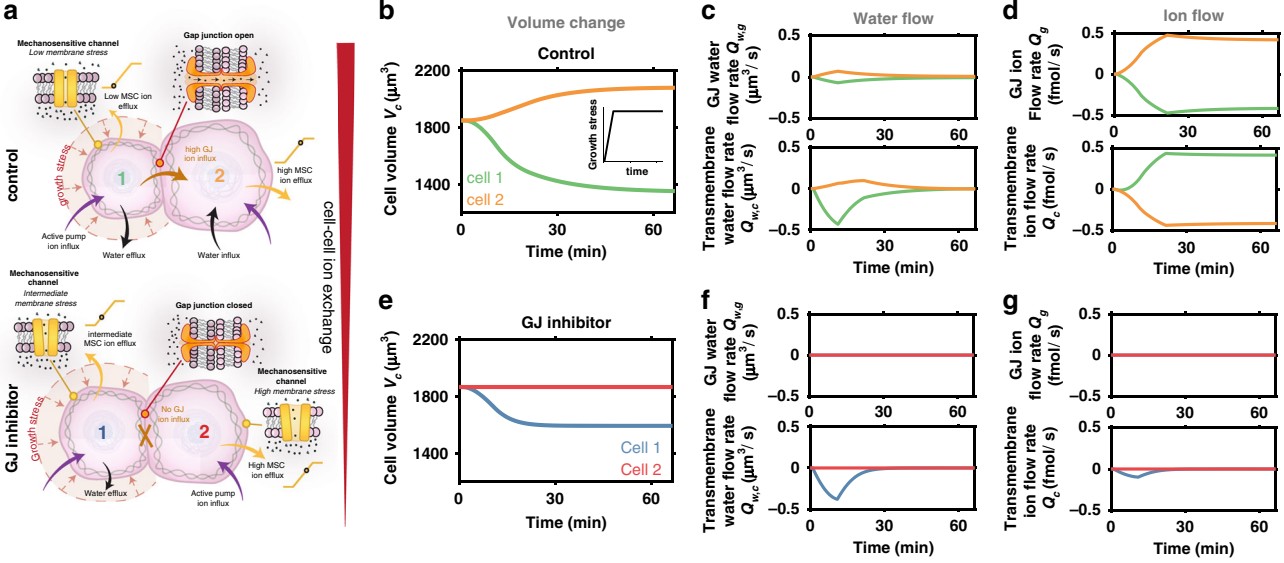

**Fig. 2 Role of gap junctions (GJs) in cellular volume control. a** Schematic of solid growth stress driving swelling in neighboring cell which is prevented by inhibition of ion transport across GJs. **b** In the control case, the loaded cell shrinks and the neighboring cell swells. Inset shows the loading profile that increases over time. **c** Control case water flow rate $Q_w = A J_v$ across GJs and the membrane. **d** Control case ion flow rate $Q = A\dot{n}$ across GJs and the membrane. **e** When ion transport across GJs is inhibited ($\omega_g = L_{p,g} = 0$), there is negligible volume change in the neighboring cell. **f** Inhibition case water flow rate $Q_w = A J_w$ across GJs and the membrane. **g** Inhibition case ion flow rate $Q = A J$ across GJs and the membrane.

for the cell to maintain a high osmotic pressure, allowing it to retain water. When solid stress on the cell's surface increases, its internal hydrostatic pressure also increases (Supplementary Fig. 1b) and water is squeezed from the cell (Fig. 2c); recall that water flow is partly driven hydrostatic pressure differences (Eq. 4). The loss of water relieves tension in the cell's membrane, thereby reducing the permeability of its MS channels (via Eq. 5). However, as ion pumping remains active, there is a continuous intake of ions (Supplementary Fig. 1f). Overall, the cell loses fewer ions, but the rate of ions entering the cell remains relatively constant and therefore there is a net increase in its ion concentration. This increases the loaded cell's osmotic and hydrostatic pressure (Supplementary Fig. 1b, c) relative to its connected neighbor, and the differences generate a flow of ions through their GJs from the loaded to the unloaded cell (Fig. 2d). The resulting increase in the unloaded cell's ion concentration causes it to absorb additional water from the external media and swell (Fig. 2b). Although its MS channels are now more permeable due to increased membrane tension, the constant flow of ions from the loaded cell maintains the neighbors swollen state. Thus, compression of a cell increases its osmotic pressure relative to its connected neighbor due to closing of MS channels. This drives a flow of ions across GJs into the unloaded cell, causing it to absorb water and swell.

To further analyze the role of cell–cell transport in volume regulation, we inhibit GJs by reducing their ion and water permeability (via $\omega_g$ and $L_{p.g}$, respectively) to zero. The loaded cell shrinks (Fig. 2e), but it loses less water than it would under control conditions. This reduction in volume again relieves membrane tension, reducing the permeability of MS channels and increasing the cell's overall ion concentration. However, as GJs are blocked there is no loss of ions to the neighboring cell (Fig. 2g) and thus the loaded cell sustains its high osmotic pressure (Supplementary Fig. 2c). This allows it to retain more water (and maintain a high hydrostatic pressure) to oppose the applied compressive stress. Additionally, there is no swelling predicted in the neighboring cell (Fig. 2e), owing to the lack of cell–cell ion exchange. Clearly, ion transport across cellular GJs plays an important role in cellular volume regulation, with an increasing GJ permeability driving larger differences in cell volume (Supplementary Fig. 3b). Although we assume that cells retain an approximately spherical shape, our simulations also suggest that similar trends emerge in connected elongated cells (Supplementary Fig. 4).

In this analysis, we have implicitly assumed that the external osmotic and hydrostatic pressures are spatially uniform (i.e. act equally on both connected cells). However, this may not always hold true, especially in large organoids where media perfusion can be impaired. We therefore explore the influence of spatial variance of osmotic pressure and hydrostatic pressure in Supplementary Notes 9 and 10, respectively, in the context of mechano-electro-osmotic flow. Our analyses reveal that hydrostatic pressure gradients have an impact similar to differences in applied solid stress of equivalent magnitudes (Supplementary Fig. 8c); briefly, increased external hydrostatic pressure in a loaded cell drives ions across GJs into a connected neighbor, which consequentially swells. Therefore, consideration of solid growth stress can also implicitly describe the influence of variations in external fluid pressure. Conversely, equivalent differences in osmotic pressure have a markedly lower influence on cellular volumes due to the ability of cells to adapt in response to osmotic shocks (Supplementary Fig. 7). In the next section, we proceed to generalize our analysis of volume changes in a two-cell system to a multicellular cluster.

**Intercellular ion transport drives spatial variations in cell volume within proliferating solid tumors.** Multicellular spheroids are an increasingly promising experimental model for cancer development, that aim to bridge the gap between in vitro and in vivo conditions[11]. In such 3D environments, however, it remains unclear how individual cells regulate their volumes and coordinate to advance tumor progression and matrix invasion. Therefore, we next consider how our mathematical framework can be extended to predict fluid and ion exchange within connected cells in a tumor organoid (as driven by differences in solid stress, hydrostatic pressure, and osmotic pressure). While our current model can readily be adapted to simulate a series of discrete connected cells, more physical insights can be gained from a continuum formulation that describes cellular behavior within a spherical organoid (Fig. 3a). Continuity requires that for any cell within a cluster, the change in its number of ions must equal the amount gained and lost through GJs to neighbors and through the cell membrane. First, considering cell–cell fluid exchange, recall that the volume flux across GJs between two connected cells depends on their hydrostatic pressure difference, with $J_{v,g,i} = -L_{p.g}(P_i - P_{i+1})$. The gradient of hydrostatic pressure between these cells may be written as $\nabla(\Delta P) = (\Delta P_i - \Delta P_{i+1})/r_0$. For a cell within a longer series, this can be extended to develop an expression for the Laplacian of the hydrostatic pressure, whereby:

$$r_0^2 \nabla^2(\Delta P) \approx \Delta P_{i-1} + \Delta P_{i+1} - 2\Delta P_i. \qquad (7)$$

In a spherical coordinate system, assuming circumferential symmetry, this Laplacian can be rephrased as $r_0^2 \nabla^2(\Delta P) = \frac{r_0^2}{r^2} \frac{\partial}{\partial r}\left(r^2 \frac{\partial(\Delta P(r))}{\partial r}\right)$, where $r$ is the radial position in a spherical organoid. Further, the volume flux across GJs may also be extended to describe a cell in-series and connected on both sides, such that $J_{v,g,i} = -L_{p.g}((\Delta P_i - \Delta P_{i-1}) + (\Delta P_i - \Delta P_{i+1}))$. Altogether, we can then enforce continuity to formulate a continuum expression that describes cellular volume at position $r$ within a multicellular spheroid, such that

$$\frac{\partial V_c(r)}{\partial t} = A_g L_{p.g} \frac{r_0^2}{r^2} \frac{\partial}{\partial r}\left(r^2 \frac{\partial(\Delta P(r))}{\partial r}\right) - 4\pi r_c^2(r) L_{p.m}(\Delta P(r) - \Delta \Pi(r)), \quad (8)$$

where the first term on the right describes the volume change driven by fluid flow through GJs in a cell at position $r$, and the second term accounts for cellular exchange of fluid with the surrounding media. Similarly, the number of ions entering or leaving a cell may be expressed by

$$\frac{\partial N(r)}{\partial t} = A_g \frac{r_0^2}{r^2}\left[\frac{\Pi^{ext}}{RT} L_{p.g} \frac{\partial}{\partial r}\left(r^2 \frac{\partial \Delta P(r)}{\partial r}\right) + \omega_g \frac{\partial}{\partial r}\left(r^2 \frac{\partial \Delta \Pi(r)}{\partial r}\right)\right] - 4\pi r_c^2(r)((\omega_{ms}(\sigma(r)) + \omega_l + \gamma)\Delta \Pi(r) - \gamma \Delta \Pi_c),$$
$$(9)$$

where the first term on the right describes the ions gained and lost through GJs in a cell at position $r$ (following from Eq. 6), and the second term accounts for cellular exchange of ions with the surrounding media via pumps and channels. We solve our system of equations at steady state (i.e. $\frac{\partial N(r)}{\partial t} = \frac{\partial V_c(r)}{\partial t} = 0$) using multiphysics software COMSOL to simulate local cell behavior within a spherical organoid of radius $r_{max}$ (see 'Methods for more details). As the GJ flux vanishes at the cluster boundary, a zero-flux condition is enforced on the spheroid surface.

Proliferation of cells within a growing cluster generates solid compressive stresses, additionally compounded by matrix stretch and cell confinement. Interestingly, it has been shown that such local compressive stresses are spatially non-uniform across the cancerous structure[46,47], frequently highest at the cluster core and

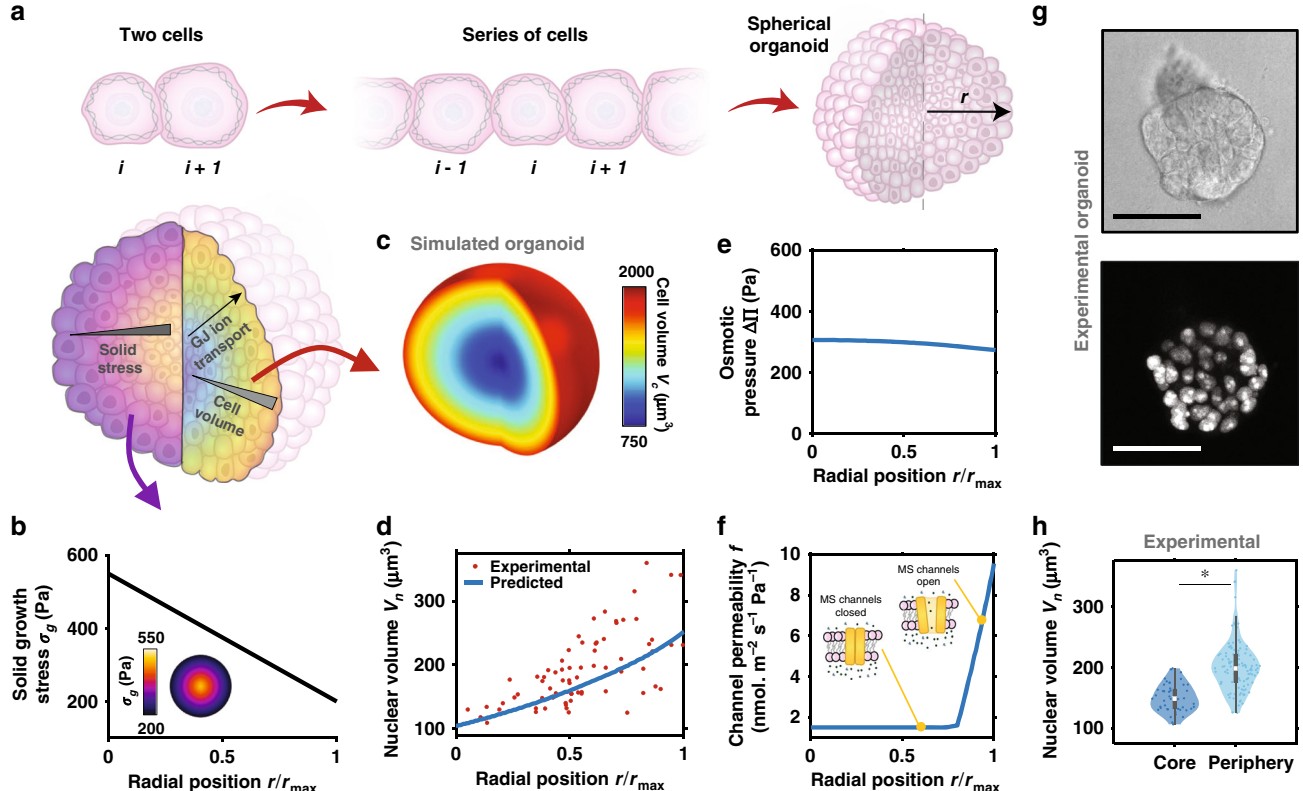

**Fig. 3 Spatially non-uniform cell volume in cancer organoid. a** Schematic of model extension to 3D spherical organoid. **b** Applied solid growth stress $\sigma_g(r)$ is highest at the core and spatially non-uniform. **c** Predicted spatial cell volume. **d** Predicted and experimental (day 5) spatial nuclear volume in the organoid. Predicted spatial **e** osmotic pressure $\Delta\Pi$ and **f** channel permeability $f = \omega_{ms}(\sigma) + \omega_l$. **g** Cross-section images of epithelial cancer organoid developed from MCF10A cells at day 5 of growth with GFP-NLS-labeled cell nuclei. Scale bar 50 μm. **h** Nuclear volume of cells at organoid core and periphery at day 5 of growth ($n = 5$ multi-cellular clusters over three independent experiments). The boxes represent the interquartile range between the first and third quartiles, whereas the whiskers represent the 95% and 5% values, the squares represent the median, and the shaded region bounds indicate the maxima and minima. A two-tailed Student's $t$-test was used when comparing the difference between two groups *$P < 0.001$.

lowest at the periphery, with maximum stress $\sigma_g^{max}$. We consider this compressive stress $\sigma_g(r)$ to act uniformly on the surface of a cell located at position $r$ in the spheroid. To estimate the distribution of compressive solid stresses in our multicellular clusters, we simulate spheroid growth in a hydrogel system (Supplementary Note 12). Simulations suggest that the solid stress varies from ~550 Pa at the core to ~200 Pa at the periphery, which we apply in our analysis (Fig. 3b). Our model predicts that cell volume is lowest at the core ($V_c(r = 0) = 750\ \mu m^3$) and increases radially ($V_c(r = r_{max}) = 2000\ \mu m^3$), as shown in the organoid contour plot in Fig. 3c. We have shown that the ratio of nuclear to cell volume remains constant during volume changes[12], maintaining a value of $V_n : V_c \cong 0.14$. Applying this ratio, we can predict nuclear volumes across the organoid (Fig. 3d), and show strong agreement with our recent experiments[12] as discussed in more detail in the next section. At the core, where there is the most significant cell shrinkage, water loss reduces the membrane tension and thus MS channel permeability is impaired (Fig. 3f). As a result, the osmotic pressure of internal cells increases (Fig. 3e). Radially, the solid growth stress reduces, and ion channels become increasingly permeable provided the membrane stress exceeds the critical value $\sigma_c$ (Eq. 5). Within the whole organoid, this generates an intercellular ion concentration gradient that propagates radially (Fig. 3e), driving an ion flow that increases the osmotic pressure in peripheral cells and causes them to absorb water. The loss of ions from core cells impedes their ability to retain water, and thus they are highly compressed

during loading. As such, at steady state there is a balance attained between the solid growth stress (that drives a water efflux) and cellular ion concentrations (that drive a water influx). During an early stage of organoid growth when solid stress is low and approximately uniform, there is no predicted spatial variation in cell volume (Supplementary Fig. 10).

Further, in recent work[12], we obtained grade2 ER+ invasive ductal carcinoma breast cancer samples from a human patient, which were fixed, sectioned, and stained for imaging with confocal microscopy (Supplementary Fig. 11a). Within the tumor mass, spheroidal acinar-like clusters of cells surrounded by basement membrane were identified, which share characteristics with the 3D experimental model data reported in this study. In these acinar-like structures, nuclear volumes were observed to increase from core to periphery, consistent with our simulations (Supplementary Fig. 11b). In summary, our model suggests that ion flow through GJs, as generated by differences in cellular ion concentrations, is a significant driving factor in cell swelling and shrinkage within multicellular organoids. In the next section, we discuss additional recent experiments to validate model predictions.

**Experimental evidence validates model predictions that GJs mediate cell swelling in multicellular spheroids.** In recent work, we experimentally uncovered that locality within a breast cancer organoid governs cell volume and stiffness[12]. We seeded single MCF10A human breast epithelial cells into 3D hydrogels

composed of 4 mg/ml Matrigel and 5 mg/ml alginate (see Methods for details), such that the gels had a shear modulus of approximately 300 Pa to reflect the environment of in vivo breast carcinoma[48]. Initially, an isolated cell proliferated to form a spherical cluster (day 3), continuing to grow into a larger spheroid (day 5) with cells present both in the core and at the periphery (Fig. 3g). Further growth led to invasive branches extending into the surrounding matrix. Cells were transfected with a green fluorescent protein (GFP) tagged nuclear-localization signal (NLS), which enabled measurement of nuclear volume using 3D confocal microscopy (Fig. 3g). Importantly, we identified that the ratio of nucleus to cell volume remained constant over a wide range of organoid sizes and cell positions[12], in agreement with previous findings[49,50], which allowed us to measure nuclear volume in lieu of cell volume. At an early stage of growth, all cells had similar nuclear volumes (Supplementary Fig. 10e). However, as the organoid further developed, nuclear volume began to correlate strongly with cell position within the cluster. Cells toward the core (inner 40% of organoid radius) were significantly smaller than those in peripheral regions (Fig. 3h), with a volume range in strong agreement with our model predictions (Fig. 3d). Interestingly, we also demonstrated that a reduction in volume was highly correlated with an increase in cell stiffness (measured using optical tweezers active microrheology[51]), most likely due to increased molecular crowding[49,52]. These experimental results both validate our model findings and highlight the importance of understanding the mechanisms of volume change within multicellular spheroids.

As our model suggests that GJs are a critical mediator of spatial volume variation within breast cancer organoids, we next explore how cell behavior changes when GJs are blocked. The spatial variation in cell volume is shown to decrease significantly (Fig. 4a), ranging from ~1500 μm³ at the core to ~1550 μm³ at the periphery. As per the control case (active GJs), compressive stress reduces cell volume throughout the spheroid. However, when the MS channels of inner cells close (Fig. 4d), there is no loss of ions through GJs to relieve the cells' high ion concentrations. Thus, relative to the control case, cellular osmotic and hydrostatic pressure are significantly higher at the organoid

core (Fig. 4e). Further, peripheral cells have a low osmotic pressure because they do not gain ions from their neighbors; therefore they do not absorb water and swell. In fact, they retain a volume similar to that during an early stage of organoid growth (Supplementary Fig. 10e). In our experiments, we inhibited GJs by adding 500 μM carbenoxolone to the organoid-matrix system on day 3 (before a volume gradient was present)[12]. In agreement with our simulations, on day 5 we did not observe significant spatial differences in nuclear volume (Fig. 4a, c). This further supports our model findings that ion diffusion driven by an intercellular osmotic pressure gradient is the critical factor in cell swelling and shrinkage within the organoid.

Finally, as cell volume clearly additionally depends on the solid stress gradient, we examined the influence of reducing the solid stress acting on the cell cluster. Our model predicts that a reduction in the maximum (core) solid stress ($\sigma_g^{max} = 400$ Pa) also reduces the spatial variation in cell volume (Fig. 4b). The reduction in stress allows more cells to sustain their MS channel permeability (Fig. 4d), permitting a loss of ions to the interstitium and thereby lowering their osmotic pressure. This reduces intercellular ion diffusion and thus lowers cell swelling in peripheral regions of the cluster. We performed additional experiments that can test these predictions; organoids were cultured in a collagen (3.5 mg/ml)/Matrigel (0.5 mg/ml) matrix to achieve a day-5 nucleus volume distribution similar to the Matrigel/alginate system (Supplementary Fig. 12). We then reduced solid stress on day 5 by degrading collagen fibers with collagenase. After 6 h, we observed a significant increase in nuclear volume at the core and a reduction at the periphery (Fig. 4b, c); the volume gradient became weaker, in agreement with model findings. In summary, the spatial variation in cell volume within the breast cancer organoid may be understood to depend on intercellular ion diffusion in response to an osmotic pressure gradient, as driven by non-uniform external solid stress.

## Discussion

In this study, we propose a mechano-osmotic model to investigate how cell volume is regulated within multicellular systems.

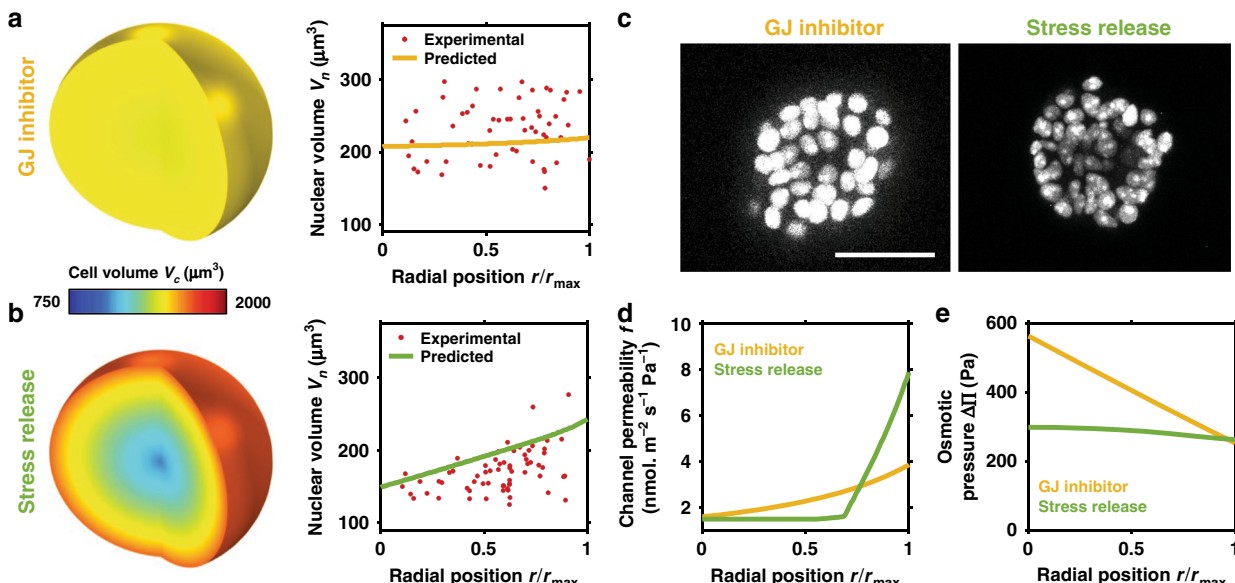

**Fig. 4 Influence of GJ inhibition and stress release on cellular swelling.** Predicted and experimental (day 5) spatial cell and nuclear volumes under inhibited gap junction (**a**) and stress release (**b**) conditions in the organoid. **c** Cross-section images of GFP-NLS-labeled MCF10A cells at day 5 ($n > 3$). Scale bar 50 μm. Predicted spatial **d** channel permeability $f = \omega_{ms}(\sigma) + \omega_l$ and **e** osmotic pressure $\Delta\Pi$.

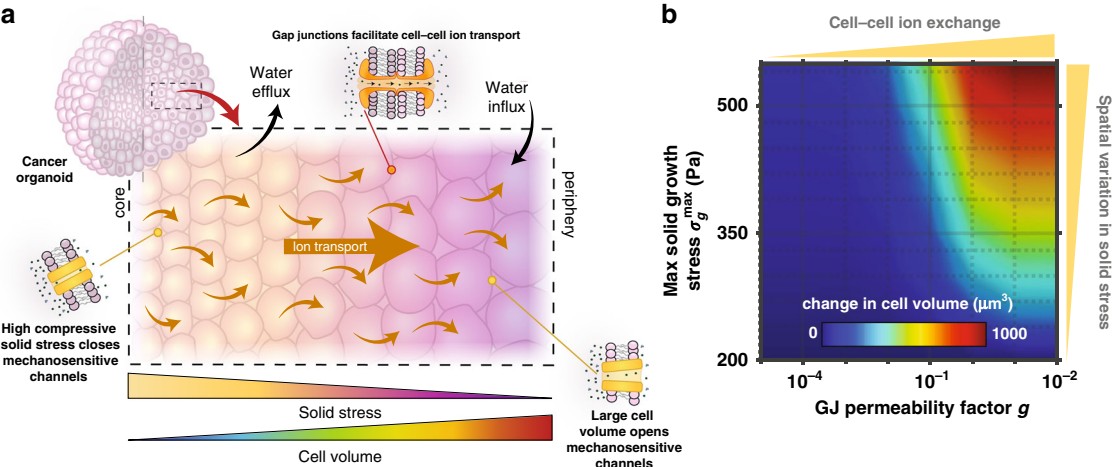

**Fig. 5 Gap junctions mediate spatial volume differences. a** Non-uniform solid growth stress drives spatial variation in cell volume via gap junction-mediated ion flow. **b** Spatial variation in cell volume is increased both by high cell–cell ion transport and high solid growth stress. GJ permeability factor $g$ scales the GJ permeability with $\omega'_g = g\omega_g$ and $L'_{p.g} = gL_{p.g}$.

We hypothesized that GJs could play a key role in the volume dynamics of connected cells, by mediating solute flow as driven by cell deformation. Volume control depends on an interplay between multiple cellular constituents, including GJs, MS ion channels, energy-consuming ion pumps, and the actomyosin cortex, that coordinate to manipulate cellular osmolarity. In connected cells, our model suggests that mechanical loading significantly affects how these components cooperate to transport ions, and precise volume control is impacted by the evolution of osmotic pressure gradients between cells. Combining the modeling framework with our recent experiments, we ultimately identify that GJs could amplify spatial variations in cell volume within multicellular spheroids and, further, suggest how the process depends on proliferation-induced solid stress (Fig. 5). Initially considering a simple two-cell system, our model predicts that compressive stresses squeeze water from a cell, and a subsequent reduction in membrane tension and loss of MS channel permeability impedes the flow of ions to the external media. As ion pumps continue to transport new ions into the cell, there is an effectual increase in the cytosolic ion concentration. In connected cells, a flow of ions through GJs is thus generated from the loaded cell, thereby increasing the osmolarity of its neighbor and causing it to swell. This proposed mechanism is supported by the blocking of GJs, which is revealed to prevent such volume changes and, also, reduce shrinkage of the loaded cell.

We next extended our framework to explore how spatial variations in cell volume can emerge within a multicellular tumor. Cancer cells are typically situated within a confining matrix, and as the cells proliferate they both push against their neighbors and displace the elastic matrix. Associated solid stresses develop within the cluster, generally evolving to be highest at the cluster core[46,47]. In response, core cells tend to be significantly compressed and lose volume. Our model indicates that this can relieve cellular membrane tension, leading to the closure of MS ion channels. A reduction in this channel permeability would dictate that fewer ions are lost to the interstitium. Overall, as ions are still pumped into the cells via active transport, there is a net increase in their internal ion concentration. With a radial reduction in solid stress loading, our simulations suggest that more ion channels remain open toward the organoid periphery, allowing peripheral cells to maintain lower osmotic pressures. Following the mechanisms highlighted by our two-cell analysis, this could

drive a radial intercellular flow of ions from cells in the core, thereby increasing the ion concentration in cells situated toward the periphery. These cells then swell in response to a water influx driven by their increased osmolarity (Fig. 5a). Interestingly, from our continuum expressions (Eq. 8, 9), we can identify an effective length scale for cell volume changes in response to solid stress $L = \frac{r_0}{r_c}\sqrt{\frac{\omega_g A_g}{4\pi(\omega_{ms}(\sigma) + \omega_1 + \gamma)}}$, which reveals that the transmission distance increases with either increasing GJ solute permeability $\omega_g$ or reduced MS channel permeability $\omega_{ms}(\sigma)$. Further, we can readily obtain analytical solutions for our continuum formulations at the limits of GJ permeability (Supplementary Note 15), to highlight that GJs can amplify spatial differences in cell volume. In fact, our simulations clearly indicate that increasing GJ permeability promotes larger volume differences across the multicellular spheroid (Fig. 5b). Spatial variation in cell volume is also amplified at higher solid stress gradients.

GJs play a critical role in supporting many physiological operations, including embryonic development[53] and collective cell migration[54]. Clearly, our framework could be used to analyze such biological systems and provide mechanistic insight into their dependence on ion transport and differential cell swelling. Future advancements should also focus on detailing the interdependence between dynamic actomyosin contractility and cell osmolarity, building on our previous work to further understand how the two-way feedback between stress and signaling guides nuclear gene expression[55], dynamic force generation[56], and cancer invasion[14]. Here we have implicitly considered small unified cell clusters. Future model implementations should analyze the influence of spatial variations in cell characteristics (stiffness, connectivity) associated with hypoxic conditions[57] and the evolution of cell volume during invasion (cell separation and loss of GJs). Importantly, in our main analysis, we limit ourselves to the consideration of a single ion species and the associated channels and pumps. However, the influence of multiple charged species, membrane potential, and electroneutrality is explored in Supplementary Note 8, with our analysis suggesting that the reduced single-species framework captures key trends predicted by our full mechano-electro-osmotic model.

In cancer progression, the precise influence of GJs remains a point of avid debate[58]. In early studies, loss of intercellular communication was identified to be characteristic of cancer cells[59]. However, it has been reported that low expression of the

GJ protein connexin correlates with both positive and negative prognoses across a range of cancer types[60,61]. Particularly in late stage tumors, there is an increasing body of evidence that indicates expression of connexin is associated with tumor malignancy, growth, and invasion[62–64]. In this study, we identified that ion flow through GJs promotes peripheral cell swelling in loaded breast cancer model. Cell progression through the cell cycle is reportedly dependent on cell size, though the mechanisms are not yet clear[65,66]. Future implementations should consider the feedback between mitosis, cell size, solute/solvent flow, and force balance, building on previous models for spheroid growth[22,67]. Controlled swelling has been shown to increase cell proliferation[19,20], which could hint at a mechanism by which GJs support cancer progression. Further, in our recent experiments, we demonstrated that swelling correlated with reduced cell stiffness[12], which has been suggested as a metastatic biomarker associated with increased invasive potential[68]. Accordingly, we found that osmotic swelling increased cell invasiveness, while blocking GJs led to a reduction in the number of invading branches, indicating that GJ-mediated swelling promotes matrix invasion. In summary, our findings suggest that intercellular ion flow may be an important mediator of breast cancer progression. Future studies should aim to characterize this behavior across alternative cancer cell lines to assess if these results are generalizable to other types of cancer. Our proposed model could also be extended to aid the development of therapeutics that target inter- and extra-cellular ion transport.

## Methods

**Simulation procedure**. For the two-cell dynamic analysis, the model was implemented using an ODE solver (ode23s) in MATLAB (v. 2019a, MathWorks). Initial conditions were identified by solving Eqs. 1–6 at steady state. Cell behavior was simulated over 60 min, with a time-dependent load $\sigma_{g,1}$ introduced following 1 min (Supplementary Fig. 1a). All material parameters are summarized in Table S1. For the cell cluster analysis, a 2D axisymmetric spheroid model of radius $r_{max}$ was constructed in multi-physics software COMSOL (v. 5.4, COMSOL AB). Using in-built PDE solver functionality, Eqs. 8, 9 were solved in conjunction with mechanical equilibrium (Eq. 3) to determine spatial steady-state cell behavior in response to an applied non-uniform load $\sigma_g(r)$. For the control case, $\sigma_g^{max} = 550$ Pa and $\sigma_g^{min} = 200$ Pa. For simulations of stress-release, the maximum stress was reduced to $\sigma_g^{max} = 400$ Pa. A zero-flux condition was enforced on the spheroid surface.

**Cell lines and cell culture**. MCF10A cells (ATCC, CRL-10317) were cultured in complete medium at 37 °C with 5% $CO_2$. The complete medium is made of DMEM/F12 medium (Invitrogen, 11965-118) supplemented with 5% horse serum (Invitrogen, 16050-122), 20 ng/ml epidermal growth factor (Peprotech, AF-100-15), 0.5 µg/ml hydrocortisone (Sigma, H-0888), 100 ng/ml cholera toxin (Sigma, C-8052), 10 µg/ml insulin (Sigma, I-1882), and 1% penicillin and streptomycin (Thermo Fisher, 15140122). We transfected the MCF10A cell line with GFP-NLS lentivirus to visualize the cell nucleus following the product manual (Essen Bioscience, 4475), and the stable cell line was maintained in T-25 cell culture flask with complete medium and 0.4 mg/ml puromycin (Thermo Fisher, A1113802). Subculture was performed when cells grow into 80% confluency. Briefly, cells were washed with PBS three times before 1 ml of 0.05% trypsin-EDTA solution (Thermo Fisher, 25300054) was added. Then the T-25 flask was incubated at 37 °C for 15 min. After most of the cells detached from the flask, cells were collected, centrifuged (180 × g, 5 min), and resuspended into a new flask.

**Growth of MCF10A clusters**. The MCF10A clusters with invasive phenotype were cultured and induced following previously established protocols[69,70]. Briefly, two hydrogel systems were used in this study for 3D cell culture, including Matrigel/alginate hydrogel and collagen/Matrigel hydrogel. For the Matrigel/alginate hydrogel, cells were mixed with alginate (FMC Biopolymer), calcium sulfate (Sigma, 255696), and Matrigel (Corning, 354234) to form gel precursor solution with final concentrations of 5 mg/ml, 20 mM, and 4 mg/ml, respectively. For collagen/Matrigel hydrogel, cells were mixed collagen (Advanced BioMatrix, 5133) and Matrigel with final concentrations of 3.5 mg/ml and 0.5 mg/ml, respectively. The gel precursor solution was then incubated in 37 °C for 30 min to form cell-laden hydrogel and cultured in complete cell culture medium for 10 days. To inhibit GJs, 500 µM carbenoxolone was added to the complete cell culture medium. To reduce solid stress within the clusters in collagen/Matrigel system, collagenase D

(Sigma, 11088866001) was used to remove the matrix. For preparation of human tissue samples, the reader is referred to our prior work[12].

**Nuclear volume measurements**. The 3D structure of the MCF10A clusters was imaged with a confocal microscopy (Leica, TCL SP8), and deconvolution (HUYGENS software) was applied to the image to improve the z resolution of traditional confocal microscopy. The volume measurements were repeated using a super-resolution microscopy (stimulated emission depletion (STED) microscopy) with a spatial resolution of ~100 nm. A consistent volume pattern was observed, as shown in Supplementary Fig. 9. The nuclear volume was then calculated by the number of voxels contained within the nuclear structures using a customized algorithm in MATLAB (v. 2017a, Mathworks).

**Statistics and reproducibility**. A two-tailed Student's t-test was used when comparing the difference between two groups. In the box plots, the boxes represent the interquartile range between the first and third quartiles, whereas the whiskers represent the 95% and 5% values, and the squares represent the median. Five to fifteen multi-cellular clusters were measured in each experiment, and all measurements were performed in at least three independent experiments to verify the reproducibility of the experimental findings.

**Reporting summary**. Further information on research design is available in the Nature Research Reporting Summary linked to this article.

## Data availability

Data supporting the findings of this study are available within the article, Supplementary Information and Source Data, and are available from the corresponding author on request. Source data are provided with this paper.

## Code availability

The MATLAB and COMSOL files used for this research are openly available on Github (https://github.com/EoinMcEvoy).

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

## Acknowledgements

This work was supported by National Cancer Institute Awards U01CA202177 (V.B.S.), U54CA193417 (V.B.S.), R01CA232256 (V.B.S.), and U01CA202123 (M.G.); National Institute of Biomedical Imaging and Bioengineering Awards R01EB017753 (V.B.S.) and R01EB030876 (V.B.S.); NSF Center for Engineering Mechanobiology Grant CMMI-154857 (V.B.S.); NSF Grants MRSEC/DMR-1720530 (V.B.S.) and DMS-1953572 (V.B.S.); Alfred Sloan Research Fellowship (M.G.).

## Author contributions

E.Mc. and V.B.S. designed the theoretical models and carried out the computations; Y.H. and M.G. designed and conducted the experiments; E.Mc., Y.H., M.G., and V.B.S. analyzed and interpreted the data; E.Mc., Y.H., M.G., and V.B.S. prepared and edited the manuscript.

## Competing interests

The authors declare no competing interests.
