## [Peer Review File · Nature Communications]

REVIEWER COMMENTS

Reviewer #1 (Remarks to the Author):

This is a very well-presented study in which the authors propose a mechano-osmotic model to investigate the spatial distribution of cell volume within multicellular spheroids. The proposed formulation allows the volume change to occur via fluid movement and ion exchange between the adjacent cells (through the gap junctions, GJs), or between a cell and extracellular environment (i.e. mechanosensitive channels, leak channels, and ion transporters). To verify the model, the authors use three sets of experimental data on - a MCF10A spheroid as the control – a spheroid after inhibiting GJs -a spheroid after releasing solid stress by degrading the surrounding hydrogel. Finally, the authors concluded that GJs play an important role in the spatial variation of cell volume within spheroids.

While the study is novel, the model constituents have been selected wisely and the governing physics has been well explained, there are some concerns that need to be addressed before further consideration for publication:

-Conclusions in this study are drawn based on data obtained from only one breast cancer cell type. Therefore, the question is if the conclusions are generalizable? To address this question the authors are encouraged to repeat the experiments for one or two more cancer cell lines to evaluate whether GJs are important in the volume control of the other cell types. If the experimental data for different cancer cell types are available, the authors can explore how the model's parameter would change for the other cell types (if any change in parameters is necessary). On the other hand, since the motivation of the paper is on developing a novel model, the authors may tone-down the conclusions and modify the presentation of the paper to emphasis on the model.

-Considering the importance of parameters in predicting the volume of cells in a spheroid, calibration procedures need to be more dissected. All 16 parameters of the model have clear physical meanings and most of them can be estimated directly from experimental tests. However, to evaluate some, curve fitting techniques might be required. It is suggested that for the former set of parameters the authors provide more details about the experimental procedure (especially for μ and ωg) and for the latter set of parameters, they evaluate the sensitivity of the model to an alteration in input parameters.

-Particularly, since 16 parameters are involved in the simulations, it would be very beneficial to obtain some relevant phase maps to evaluate the sensitivity of the important outputs such as cell volume to the simulation parameters.

-The proposed model has been verified using experimental data. Although the model provides an appropriate description of the mechanisms affecting the observed behaviour, it cannot be used to verify the experiments. Therefore, it is highly recommended that the authors repeat the experiments for each condition, including control, GJ inhibition, and stress release tests (especially control and GJ inhibition tests), at least three times to obtain statistically reliable data. If this has been already done, the number of repeats and spheroids that have been analysed needs to be reported.

-Why the hydrogel used for the control (4mg/ml Matrigel + 5 mg/ml alginate) is different from that in stress release experiment (0.5mg/ml Matrigel + 3.5 mg/ml collagen)? Would a different gel affect the model's parameters (the ones related to the interplay between the cell and extracellular environment)? Have the authors used the same set of parameters for these two cases? Additionally, it is suggested that the authors provide the nuclear volume versus radial position plot for the stress-release case before unloading and compare it with the control test (I guess Fig. S9B and Fig. 3D are the same, are the data of stress release test before unloading included in the plot?).

-Figure 3D shows that the nuclear volume varies within a narrow range (between 100 and 300 μm^3). Considering the fact that the nuclear volume is very sensitive to the quality of the images (assuming that

the nuclei are spherical, changing the radius from 3 to 4 μm would cause a change in volume from 113 to 268 μm^3), it is suggested that the authors provide some information about the degree of accuracy of the method used to estimate the nuclear volume from the confocal images treated with the deconvolution software.

-What is the role of the spheroid's initial size? Would the formulation remain valid when cancer cells start leaving the tumour, or when it grows so that a hypoxic core is developed in the middle of the spheroid? Or the formulations are valid only when the size of the spheroid is smaller than a specific value?

-The effect of hydrostatic pressure perturbation (dP_{ext}) was shown to be negligible when only two cells were modelled. Could this be generalized to 3D (as shown in ref [11])? Furthermore, is it possible that the osmotic pressure perturbations have some effects on the results? Although, from the mathematical point of view, considering Eq. 7, the probable effects should be the same as that for hydrostatic pressure perturbation.

-In Eq. 4, the cell was assumed to be a sphere with a radius of r_c . After converting the discrete model to a continuum formulation, r_c still appears in Eq 7. Have the authors considered it as a function of cell volume (V_c)? What would happen if the authors consider a different shape for the cells rather spheres?

-Please double-check Eq.9. I guess the right-hand side should be $\Delta\Pi_{i-1} + \Delta\Pi_{i+1} - 2*\Delta\Pi_i$

Reviewer #2 (Remarks to the Author):

The paper by McEvoy et al entitled Gap Junctions Amplify Spatial Variations in Cell Volume in Proliferating Solid Tumors develops a model describing the role of ion and water permeation across gap junctions to regulate cell volume in a cancer aggregate submitted to internal and external mechanical constraints. The model implements, from first principles, the multiple sources of flux of ions and water fluxes across various channel and pump types. It considers passive transports across aquaporins, ion channels and other mechanosensitive channels of ion and water between the cell cytoplasm and the intercellular gap. Active transport is also considered. Transport across the gap junction that directly relates the cytoplasm between two cells is treated individually. The mechanical stability of the cells is also modeled. The model is first worked out in a cell doublet. It is then further turned into a continuum expression that allows the authors to predict the radially averaged behavior of cell volume in an aggregate.

The conclusions of the model are strengthened by simple but reliable experimental measurements of the role of gap junctions in maintaining a cell volume gradient in the aggregate.

Overall the paper is clearly written, the calculations look sound to the best of my knowledge, the fit with the experimental data are satisfactory. Overall I think the work is of interest. I think it could benefit from some rewriting and clarifications in certain parts.

I have some suggestions regarding the model and the experimental parts.

My main suggestion is that the hypothesis of the model should be better explained in the main text or in the supplementary material.

1- In the case of passive transport across the channels (aquaporins, ion channels and gap junctions) the authors consider only the terms linked to the difference of hydrostatic and osmotic pressure. This term originates as a simplification of Onsager gradients of the chemical potentials. It does not account for advective transport. In the case of aquaporins and ion channels the assumption is common. Gap junctions have diameters that are about ten times the size of a water molecule and similar to pores created by toxins (eg hemolysin). In such cases the role of water advection and electromostic flows in ion transport of ions have been extensively addressed. Neglecting these flows is an assumption of the model, that simplifies its tractability. Since the main point of the paper is precisely the role of connexins, I

guess such hypothesis should be commented. A way to implement the advection of water in the gap junction would be to have a distinct proportionality factors relating the change in volume with osmotic pressure and hydrostatic pressure. It would amount considering a Poiseuille flow among the gap junction: eq 1 $\Rightarrow dV/dt = -Sg (\omega g P \Delta P - \omega g \Pi \Delta \Pi)$. Such an assumption have been used in a recent paper describing lumen growth in blastocyst: Chan, C.J., Costanzo, M., Ruiz-Herrero, T. et al. Hydraulic control of mammalian embryo size and cell fate. Nature 571, 112–116 (2019). <https://doi.org/10.1038/s41586-019-1309-x>

Similarly in the context of lumen growth the fluxes across channels and gaps have been worked out in the <https://www.pnas.org/content/115/21/E4751> Dasgupta et al Physics of lumen growth.

2- The assumption that the passive cortical stress is proportional to the difference of the surface to a reference surface is also a very idealized though practical assumption, assuming that cell cortex is equivalent to an elastic material. It has been used in various forms in vertex models. Here the assumption of a S_0 , to some extent, is equivalent to the assumption of a preferred volume. Cortices relax within 10 of seconds and the passive stress vanishes as fast. The assumption of an elastic shell is not uncommon but often misleading. I would suggest that the authors refer to this caveat.

3- I appreciated the table in supplementary material of the paper summarizing the numerical values that are used in the text. It refers to the authors' own measurements and to that from Jiang and Sun. It would be more informative to explain in supplementary how the values were measured in these two papers and to give a range of what the authors feel "acceptable". Which of these parameters were used to fit of the experimental data. Where does the 63.3 value comes from? Why 10% of the reference area? Is the 3 .3 really relevant?

4- Before figure 1, the sentence : We consider there to be a critical... energy input is not really clear to me. If I understand correctly you assume that the active flux of pumps is not constant but depends on the difference of the osmotic pressure to a critical value $\Delta \Pi_c$. Are there experimental evidences of this, rather than a constant active flux. If yes it may be worth mentioning.

5- It would be good to indicate in the text, how many spheroids you used to compare the model to the experiments. In figure 3 and 4, each point corresponds to one distance from the center but I guess they originate from some experimental replica.

6- The first paragraphs of the discussion until : "gap junctions play a critical..." summarizes the scenario that your model display based on the assumptions you made. It reads very much though as conclusions from experimental data that would probe the mechanism one after the other. I would rather prefer that the author clearly state that the scenario originating from the model that couple these many fluxes is compatible with experimental results showing gradient of cell volume in spheroids and there inhibition when gap junctions are altered. I suggest putting more emphasis on the fact that the model is an interpretation based on the assumptions.

I am not a great fan of the last two sentences but it is there a matter of taste.

7- Along those lines, the title should reflect more the nature of the experimental system. MCF10 aggregates are far from being perfect in vitro replica of tumors.it would be fairer in to mention say " Tumorigenic cell aggregates" or " tumor spheroids" rather than tumors.

All in all I believe the article is sounds and consistent with its assumption. It describes fairly the experimental data and is of interest to the readers of Nature Com.

Reviewer #3 (Remarks to the Author):

In this manuscript entitled "Gap Junctions Amplify Spatial Variations in Cell Volume in Proliferating Solid Tumors", McEvoy et al. consider as a building block a model of individual cell volume regulation (Ref. 26) in a multi-cellular context (first two cells and then a cell spheroid) by adding gap junctions that enable solute and solvent flows between neighbouring cells. They use the model to explain 1) how the local application of a solid stress to one cell in a doublet can lead to a differential shrinking and swelling 2) their experimental observation previously published (Ref 11) that cells in a spheroid core have a smaller nuclear volume than cell in the spheroid periphery.

The manuscript is well written and clear and the topic interesting and new but i cannot recommend its publication given some important conceptual problems in the model that i could not understand and which i detail below.

- I do not understand equation (1) of the paper. Why would this model of fluid transport hold for gap junctions ? Any soluble molecule smaller than 2 nm (i.e. ions such as Na, K, Cl) can go through a gap junction which is therefore not selective. In the absence of this selectivity assumption, how can the authors justify this formula (Kedem and Katchalsky, J Gen Physiol (1961) 45 (1): 143–179., Staverman, Trans. Faraday Soc., 1952, 48, 176.) ? Why isn't it only the difference of osmotic pressures between the non-permeable osmolites that enters the thermodynamic driving force ? Same question in equation (4), it seems to me that ion motion through selective and non-selective channels should involve different thermodynamic forces.

In my opinion, the authors should clarify the thermodynamic framework they use and also give references.

-Again equation (2) is unclear to me. Why is it the total osmotic pressure which comes as a generalized thermodynamic force and not the chemical potential of each ion specie ? Also each ion specie may have its own mobility through the gap junctions which may in particular depend of the ion charge. The authors do introduce a work of caution about this in the conclusion but this is not enough to have full trust in the model in my opinion. Again the full thermodynamic framework at the origin of these kinetic relations needs to be clarified.

-At which timescale the assumed elastic constitutive relation of the cortex is true ? The cortex turnovers through actin polymerization/depolymerization in a few minutes. At a long timescale relevant for growth, why isn't the cortical constitutive behaviour viscous ?

- The authors recall the result of Ref. 26: 'MS channels are permeable due to tension in the cell membrane, permitting a constant loss of ions to the external media (Fig S1D). However, active ion pumping ensures there is a continuous ion influx (Fig S1F) to maintain the cell's osmotic pressure higher than that of the external media, allowing it to retain water'. I think that one should be careful before including this complex model (many parameters and a non-linearity) in a larger framework for the following reasons. First, this model of cell volume regulation would need to be experimentally demonstrated. This has not been done in Ref. 26 (which is nonetheless useful from a theoretical standpoint) as in particular the cell filtration coefficient that they fit is several orders of magnitudes higher than the one reported in experiments, even considering their note in their table I. Second, the electro-static side of the problem has not been taken into account. While the number of macromolecules trapped in the cell is indeed small compared to the one of ions and does not contribute significantly to the osmotic pressure, their negative fixed charges which need to be neutralized by counter ions create an osmotic pressure that strongly affects the motion of ions (See the work of Y. Mori). This model is therefore different from the 'textbook' pump and leak model (Alberts et al., Molecular biology of the cell, see also the works of C. Peskin and Y. Mori) which minimally introduces two different ions (sodium/potassium) with a different membrane permeabilities. In this picture, the work of volume regulating pumps on the cell membrane is to exchange two ions not to simply import or export a single ion specie.

Can the authors then better relate their model of ion pumping to the actual biological way volume regulating pumps (typically the sodium/potassium pump) work ?

-The authors write 'recall that water flow is partly driven (by) hydrostatic pressure differences'. What is the magnitude of these pressures compared to the osmotic pressure variation ? A classical argument raised in this context is that a variation of 5mM of osmolarity which can be regarded as a fluctuation given that 300mM is a typical cell medium osmolarity leads to an osmotic pressure that is several orders of magnitude higher than 50 Pa. It seems difficult to reconcile this with the fact that hydrostatic pressure differences in this range would directly lead to transmembrane solvent flow that are not negligible.

Minor issues:

- 1) 'Cell proliferation is regulated by volume,...' not clear, specify which volume (cytoplasm, nucleus) and give references. There was a recent Nat.Phys. review on this topic.
- 2) 'energy consuming ion transporters,..' ion pumps would be more clear
- 3) 'ion channels on the membrane that permit passive exchange between the cytosol and extracellular fluid' is a bit misleading as the solvent does not travel through these channels but mostly through aquaporins.
- 4) 'precise control of the cytosolic ion concentration can increase or decrease cell volume' you could quote <https://doi.org/10.1016/j.bpj.2016.05.011>
- 5) 'cells at the core were highly compressed' a bit misleading again. Their volume is smaller but you have not measured if they are exposed to a larger stress than in the periphery
- 6) 'via Van't Hoffs equation...' mention that this is a dilute (and actually questionable) assumption in the cell
- 7) Why is there a dependance of $\Delta \Pi_c$ in Π_{ext} dependance ? Would an osmotic shock affect $\Delta \Pi_c$?
- 8) 'typically lying in the range of 100 – 250 Pa.' Give refs. This is highly dependent of the cell type.
- 9) In Fig 3D, when you compare the theoretical results with the experimental data, how many parameters are adjusted in the model ?
- 10) 'Interestingly, it has been shown that such stress is spatially non-uniform across the cancerous structure' which stress ? radial or hoop component ?
- 11) I find it questionable to assume a fixed spatial dependance of $\sigma_g(r)$, which i assume is the radial component of the solid stress (but what is the influence of the hoop component which cannot be zero unless the stress field is uniform ?). The solid stress, stemming from a certain growth law that needs to be specified, should be coupled via force balance in the spheroid to intra and extra-cellular water flows (work of Netti and Jain).

These flows also transport ions and above all, feedback on the stress distribution. The mechanical description of the problem at the continuum level should be made more clear.

RESPONSE TO REVIEWERS

We thank all three reviewers for their positive appraisals of our efforts to advance the current understanding of multicellular volume dynamics. We are grateful to them for providing a detailed and insightful review of our paper, which we believe has significantly improved the quality of our contribution. We have thoroughly addressed every comment/suggestion/question raised by the reviewers through significant revisions to our manuscript and we hope that our paper is now acceptable for publication in *Nature Communications*.

Reviewer #1 (Remarks to the Author):

This is a very well-presented study in which the authors propose a mechano-osmotic model to investigate the spatial distribution of cell volume within multicellular spheroids. The proposed formulation allows the volume change to occur via fluid movement and ion exchange between the adjacent cells (through the gap junctions, GJs), or between a cell and extracellular environment (i.e. mechanosensitive channels, leak channels, and ion transporters). To verify the model, the authors use three sets of experimental data on - a MCF10A spheroid as the control – a spheroid after inhibiting GJs -a spheroid after releasing solid stress by degrading the surrounding hydrogel. Finally, the authors concluded that GJs play an important role in the spatial variation of cell volume within spheroids.

While the study is novel, the model constituents have been selected wisely and the governing physics has been well explained, there are some concerns that need to be addressed before further consideration for publication:

We sincerely thank the reviewer for their detailed review. We would like to highlight the following key improvements made to the revised manuscript as motivated by the reviewer's comments:

- Analysis of additional data from a cancer patient biopsy sample and the hydrogel systems
- Detailed discussion of parameter motivation
- Phase maps for model sensitivity to key parameters
- Support for accuracy of nuclear measurements
- Expanded analysis of external hydrostatic and osmotic pressure gradients
- Analysis and discussion of alternative cell shapes
- Increased emphasis on model throughout manuscript and clarification of hypothesis

Our detailed response to specific comments follows below:

R1.1: Conclusions in this study are drawn based on data obtained from only one breast cancer cell type. Therefore, the question is if the conclusions are generalizable? To address this question the authors are encouraged to repeat the experiments for one or two more cancer cell lines to evaluate whether GJs are important in the volume control of the other cell types. If the experimental data for different cancer cell types are available, the authors can explore how the model's parameter would change for the other cell types (if any change in parameters is necessary). On the other hand, since the motivation of the paper is on developing a novel model, the authors may tone-down the conclusions and modify the presentation of the paper to emphasis on the model.

We thank the reviewer for this valid comment. Cancers are very different in different organs while the expression level of gap junctions also varies significantly in different type of cancers¹. From the experimental point of view, the volume regulation mechanism we proposed in the manuscript requires that: 1) the cells could form a three-dimensional (3D) spheroid, 2) gap junctions are formed between cells within the spheroid and 3) an unevenly distributed internal stress field emerges during spheroid

growth. The cell line we used in the manuscript is a human breast mammary epithelial cell lines (MCF10A), which is one of the most widely used cell line in 3D culture to study the tumorigenesis of breast cancer². However, to address the reviewer’s comments in-part we have performed additional analysis on cell spheroids obtained from a breast tumor patient in the revised manuscript. Please note the following additions:

“Further, in recent work³ we obtained grade2 ER+ invasive ductal carcinoma breast cancer samples from a human patient, which were fixed, sectioned and stained for imaging with confocal microscopy (Fig S11A). Within the tumor mass, spheroidal acinar clusters of cells surrounded by basement membrane were identified, which share characteristics with the 3D experimental model data reported in this study. In these acinar structures, nuclear volumes were observed to increase from core to periphery, consistent with our simulations (Fig S11B).”

Fig S11: *Characterization of cell volume heterogeneity in patient samples.* A) Schematic of a tumor biopsy from a breast cancer patient with fluorescent image showing cell in a local malignant acinus. Adapted from previous work³. Scale bar=50 μm ; B) Predicted and measured spatial cell and nuclear volumes from malignant acinus. For these simulations the reference cell volume $r_0 = 7.5 \mu\text{m}$ and the solid growth stress at the core $\sigma_g(0) = 450 \text{ Pa}$. All other parameters remain as reported for the experimental 3D model.”

Additionally, we have reframed our conclusions to specify breast cancer and modified aspects of the manuscript to emphasize the model. Further, please note the following text in the revised manuscript:

“In summary, our findings suggest that intercellular ion flow may be an important mediator of breast cancer progression. Future studies should aim to characterize this behavior across alternative cancer cell lines to assess if these results are generalizable to other types of cancers.”

R1.2: Considering the importance of parameters in predicting the volume of cells in a spheroid, calibration procedures need to be more dissected. All 16 parameters of the model have clear physical meanings and most of them can be estimated directly from experimental tests. However, to evaluate some, curve fitting techniques might be required. It is suggested that for the former set of parameters the authors provide more details about the experimental procedure (especially for μ and ω_g) and for the latter set of parameters, they evaluate the sensitivity of the model to an alteration in input parameters.

In the revised manuscript we have now provided additional details on the determination of parameters associated with gap junction permeability to fluid and ions (parameter names updated to $L_{p,g}$ and ω_g ,

respectively, in accordance with our extended model derivation as suggested by the other reviewers). Please note the following additional text in the revised manuscript:

“The density of GJs on the adhered membrane of epithelial-like cells has been estimated⁴ to lie in the range of $\rho_g \approx 2 \times 10^{14} \text{ m}^{-2}$. Assuming Poiseuille flow through GJs of radius $r_g \approx 1 \text{ nm}$ and length $l_g \approx 15 \text{ nm}$ ⁵, following Mathias *et al* (2008)⁴ and Gao *et al.* (2011)⁶, the solute permeability factor can be estimated as $L_{p,g} = \rho_g \pi r_g / (8 \eta l_g)$, where η is the viscosity of water. Within channels of radius 1 nm and at body temperature ($T = 310 \text{ K}$), the viscosity of water can be approximated as $0.5 \text{ mPa} \cdot \text{s}$ ⁷, leading to a coefficient $L_{p,g} \approx 10^{-11} \text{ m} \cdot \text{s}^{-1} \cdot \text{Pa}^{-1}$. Further, considering the molar volume of water $\bar{V}_w = 18 \text{ mL/mol}$, we can determine the solvent GJ permeability coefficient in terms of the number of moles whereby $\omega_{g,w} = L_{p,g} / \bar{V}_w = 5.82 \times 10^{-7} \text{ mol} \cdot \text{m}^{-2} \cdot \text{s}^{-1} \cdot \text{Pa}^{-1}$. Assuming similar diffusive behavior for the solutes (ions), $\omega_{g,w}$ suggests that ion coefficient ω_g should lie in the range $10^{-7} - 10^{-6}$. We determined that a value of $\omega_g = 5 \times 10^{-7} \text{ mol} \cdot \text{m}^{-2} \cdot \text{s}^{-1} \cdot \text{Pa}^{-1}$ provides good agreement between our simulated and experimentally measured volumes, assuming a cell-cell adhered membrane surface area A_g on the order of 10% of the reference cell surface.”

Further, we have also assessed the sensitivity of the model to alteration in input parameters, through the development of phase maps for cell volume. Please note the following additions to the revised manuscript:

“The influence of key model parameters on cell volume in two connected cells (in response to solid growth stress $\sigma_{g,1}$ on cell 1) is shown in Fig S3. Briefly, increasing the effective cortical compliance (via a reduction in K or h) will lead to an increase in cell volume (Fig S3A). Under loading, increasing the permeability of GJs to ions (via ω_g) will cause increased swelling of the connected neighbor (Fig S3B); this behavior also emerges from increasing the permeability to solvent (via $L_{p,g}$) due to an increase in advective flow. Increasing the threshold stress σ_c for the opening of MS channels causes cell volume to increase (due to higher ion retainment), while increasing the stress at which MS channel opening saturates σ_s will reduce cell volume (Fig S3C); this is akin to increasing channel permeability. Finally, increasing the rate of active pumping via γ will increase cell volume, while increasing the permeability of leak channels via ω_l will reduce cell volume (Fig S3D).

Figure S3: Sensitivity of model predictions to key parameters. Cell volume changes in response to an applied solid growth stress $\sigma_{g,1} = 150 \text{ Pa}$ on cell 1 associated with variance in A) cortical stiffness K

and thickness h , B) gap junction permeability to fluid $L_{p,g}$ and ions ω_g , C) mechanosensitive threshold stresses σ_c and σ_s , and D) pump coefficient γ and leak channel permeability ω_l .”

R1.3: Particularly, since 16 parameters are involved in the simulations, it would be very beneficial to obtain some relevant phase maps to evaluate the sensitivity of the important outputs such as cell volume to the simulation parameters.

Phase maps have been provided to assess sensitivity of cell volume to key model parameters, as shown in response to the previous comment (R1.2) from the reviewer.

R1.4: The proposed model has been verified using experimental data. Although the model provides an appropriate description of the mechanisms affecting the observed behaviour, it cannot be used to verify the experiments. Therefore, it is highly recommended that the authors repeat the experiments for each condition, including control, GJ inhibition, and stress release tests (especially control and GJ inhibition tests), at least three times to obtain statistically reliable data. If this has been already done, the number of repeats and spheroids that have been analysed needs to be reported.

We thank the reviewer for highlighting this point. Five to fifteen multi-cellular clusters were measured in each experiment, and all measurements were performed in at least three independent experiments to verify the reproducibility of the experimental findings. The sample number has now been reported within the methodology and relevant figure captions. Please note the following additional text in the revised manuscript:

“Five to fifteen multi-cellular clusters were measured in each experiment, and all measurements were performed in at least three independent experiments to verify the reproducibility of the experimental findings.”

R1.5: Why the hydrogel used for the control (4mg/ml Matrigel + 5 mg/ml alginate) is different from that in stress release experiment (0.5mg/ml Matrigel + 3.5 mg/ml collagen)? Would a different gel affect the model’s parameters (the ones related to the interplay between the cell and extracellular environment)? Have the authors used the same set of parameters for these two cases? Additionally, it is suggested that the authors provide the nuclear volume versus radial position plot for the stress-release case before unloading and compare it with the control test (I guess Fig. S9B and Fig. 3D are the same, are the data of stress release test before unloading included in the plot?).

These two models are both widely used in the field of breast cancer research^{8,9}. Using two extracellular matrices allowed us to demonstrate that our conclusions are more generally valid and of relevance in the field of breast cancer research. We used collagen/Matrigel system for the stress release experiments, as collagen is easily removed with collagenase, without directly impacting cell behavior. In terms of the model, for alternate gels the key parameters that would be modified are those relating to the solid growth stress experienced by the cells. These could be estimated by simulating growth in the alternate hydrogels (as per SI Section S12). In this study, the collagen/Matrigel hydrogel was designed to elicit similar spheroid growth prior to collagenase treatment as the Matrigel/alginate hydrogel, and as shown in Fig S12 the measured regional nuclear volumes are not significantly different. It was therefore not required to modify any model parameters for the Matrigel/Collagen analysis prior to stress release, which facilitated a direct comparison with our control and GJ blocking experiments. Please note the following text in the revised manuscript:

“To explore the influence of intra-spheroid stress (σ_g) we performed additional experiments whereby organoids were cultured in a collagen (3.5 mg ml^{-1}) / Matrigel (0.5 mg ml^{-1}) matrix (C/M). Cell locations were classified by position within the spheroid: inner ($r/r_{max} \leq 0.25$), inner-mid ($0.25 < r/r_{max} \leq 0.5$), outer-mid ($0.5 < r/r_{max} \leq 0.75$), and outer ($r/r_{max} \geq 0.75$). Comparison with observed day-5 nucleus volumes (Fig S12A) from the Matrigel/alginate (M/A) system revealed the volume differences were not significant ($P > 0.05$). Further, our model predictions were also found to lie within the interquartile range (between the first and third quartiles) of both systems (Fig S12B), indicating no parameter adjustment is required to capture nuclear volumes across these M/A and C/M systems.

Figure S12: A) Comparison of nuclear volume of cells at the inner, inner-mid, outer-mid, and outer organoid regions at day 5 of growth in the Matrigel/alginate (M/A) and collagen/Matrigel (C/M) systems. The boxes represent the interquartile range between the first and third quartiles, whereas the whiskers represent the 95% and 5% values the squares represent the median, and the horizontal lines show the mean. ns: $P > 0.05$; B) Predicted and experimental (day 5) spatial nuclear volumes in the M/A and C/M systems. Experimental distributions plotted at mid-point of associated range.”

R1.6: Figure 3D shows that the nuclear volume varies within a narrow range (between 100 and 300 μm^3). Considering the fact that the nuclear volume is very sensitive to the quality of the images (assuming that the nuclei are spherical, changing the radius from 3 to 4 μm would cause a change in volume from 113 to 268 μm^3), it is suggested that the authors provide some information about the degree of accuracy of the method used to estimate the nuclear volume from the confocal images treated with the deconvolution software.

We repeated our volume measurements using a super-resolution microscopy (stimulated emission depletion (STED) microscopy) with a spatial resolution of $\sim 100 \text{ nm}$. To estimate the error introduced using confocal microscopy, we imaged the same cancer spheroid with confocal microscopy and super-resolution mode in STED microscopy and compared cell volume measurement of each cell with these two methods; we found a high degree of agreement, indicating that confocal microscopy captures the nuclear volume accurately. It is worth noting that a deconvolution (using Huygens software) was applied to the data before we calculated the nuclear volume, which helps correct for the reduced z

resolution from traditional confocal microscopy. Please note the following text and figure in the revised manuscript:

“The volume measurements were repeated using a super-resolution microscopy (stimulated emission depletion (STED) microscopy) with a spatial resolution of ~100 nm. A consistent volume pattern was observed, as shown in Fig S9.”

Figure S9. Comparison of the volume measurements using stimulated emission depletion (STED) microscopy with a super-resolution mode with isotropic resolution in x, y and z, and laser scanning confocal microscopy. Nuclear volume of single MCF10A cells is measured by STED and confocal microscopy, showing consistency between two methods.

R1.7: What is the role of the spheroid's initial size? Would the formulation remain valid when cancer cells start leaving the tumour, or when it grows so that a hypoxic core is developed in the middle of the spheroid? Or the formulations are valid only when the size of the spheroid is smaller than a specific value?

The model presented in the current study may be used to analyze spheroids of any size; however, in the main investigation we assume that the external fluid and osmotic pressure is spatially uniform (gradients are explored in SI Sections S9 and S10). In larger spheroids (diameter > 300μm), external solute gradients can emerge due to limitations in perfusion and nutrient consumption, which would need to be explicitly considered¹⁰. Further, when a hypoxic core develops, cell characteristics (i.e. stiffness, GJ density) may change. Our model can readily be extended for the analysis of such systems by considering spatial variations in parameters akin to the approach taken for spatial variation in solid growth stress. In terms of cells leaving the tumor, variations in cell volume could be considered by discretely simulating individual cells within a cluster and allowing the GJ permeability factors to depend on the level and evolution of cell-connectivity (i.e. if a cell invades alone it loses GJs). Please note the following additions to the revised manuscript:

“Here we have implicitly considered small unified cell clusters; Future model implementations should analyze the influence of spatial variations in cell characteristics (stiffness, connectivity) associated with hypoxic conditions¹⁰ and the evolution of cell volume during invasion (cell separation and loss of GJs).”

R1.8: The effect of hydrostatic pressure perturbation (dPext) was shown to be negligible when only two cells were modelled. Could this be generalized to 3D (as shown in ref [11])? Furthermore, is it possible that the osmotic pressure perturbations have some effects on the results? Although, from the mathematical point of view, considering Eq. 7, the probable effects should be the same as that for hydrostatic pressure perturbation.

Hydrostatic pressure perturbations can readily be generalized to 3D by including a similar perturbation term with Eqn 9. To highlight this, we have included the following text in the revised manuscript:

“Similarly, the extension can also be incorporated to the continuum-level model via inclusion of δP^{ext} in the membrane-specific term within Eqn 9. In this instance the hydrostatic perturbation $\delta P^{ext}(r)$ would need to be spatially defined in a similar context to $\sigma_g(r)$.”

Further, it is possible to analyze the influence of osmotic pressure perturbations on spatial changes in cell volume. In response to the other reviewers, we have provided an extended supplementary analysis electrical potentials and multiple ion species, and explored external osmotic pressure gradients in this context. In brief, our model suggests that osmotic pressure gradients have a weaker influence on cell volume in comparison to solid stress or external fluid pressure of a similar magnitude, as cells can rapidly adapt to osmotic shock. Please note the following additions to the revised manuscript:

“In our analyses we implicitly assume that the extracellular ion concentrations are spatially uniform. However, within multi-cellular organoids there may also be local interstitial osmotic perturbations that can influence cell behavior. Further, in large non-vascularized cell clusters there may be an unequal distribution of external solutes. Thus, here we introduce an extension to the MEO model to facilitate exploration of such a non-uniform solute distribution on shrinkage and swelling. Recall that the water flux across the cell membrane is driven by a balance between the internal and external hydrostatic and osmotic pressures, such that $J_{v,m,i} = -L_{p,m}(\Delta P_i - \Delta \Pi_i)$, where $\Delta P_i = P_i - P^{ext}$ and $\Delta \Pi_i = \Pi_i - \Pi^{ext}$. Within the MEO model the osmotic pressure differences are stated explicitly in terms of individual species, with $\Delta \Pi_i = RT(c_{Na^+,i} + c_{K^+,i} + c_{Cl^-,i} + X/V_i - (c_{Na^+,e} + c_{K^+,e} + c_{Cl^-,e}))$. Without loss of generality this difference can be rephrased to consider a local variation in Na^+ concentration $\delta c_{Na^+,e,i}$ such that $\Delta \Pi_i = RT(c_{Na^+,i} + c_{K^+,i} + c_{Cl^-,i} + X/V_i - (c_{Na^+,e} + \delta c_{Na^+,e,i} + c_{K^+,e} + c_{Cl^-,e}))$. As an increase in the interstitial solute concentration will also affect loading on the cell membrane, the associated flux must also be updated to include $\delta c_{Na^+,e,i}$:

$$\begin{aligned} \frac{dc_{Na^+,i}}{dt} = & -\frac{A_i g_{Na}}{F V_i} \left(\phi_i - \frac{RT}{F} \ln \left(\frac{c_{Na^+,e} + \delta c_{Na^+,e,i}}{c_{Na^+,i}} \right) \right) - \frac{3pA_i}{V_i} \\ & - \frac{A_g g_g}{F V_i} \left((\phi_i - \phi_{i+1}) - \frac{RT}{F} \ln \left(\frac{c_{Na^+,i+1}}{c_{Na^+,i}} \right) \right) - \frac{S_g L_{p,g} c_{Na^+}^*}{V_i} (\Delta P_i - \Delta P_{i+1}). \end{aligned} \quad (S33)$$

Similar modifications could be made to the expressions for K^+ or Cl^- . As shown in Fig S7A, the introduction of a small osmotic shock ($5mM$) has a marked, albeit low, influence on the volume of the shocked cell. The sudden change in the external Na^+ concentration causes the osmotic pressure difference to sharply decrease, by $\delta \Delta \Pi_i = RT(5mM) \approx 13 kPa$ (Fig S7B). However, this perturbation is rapidly balanced by solvent efflux from the cell and solute influx. Overall, it is interesting to note that hydrostatic pressure change that corresponds to such an osmotic pressure fluctuation is quite low (Fig S7C), predicted to be on the order of $20 Pa$. Naturally, this magnitude depends on effective cell stiffness and resistance to volume change. The steady state volume of the connected neighboring cell is unaltered as the diffusive potential (introduced by the slight variance in the shocked cell’s solute concentration) is not sufficient to overcome the balance of electrical potentials. The same behavior may be explored with the mechano-osmotic model presented in the main paper text by similarly perturbing the external osmotic pressure Π^{ext} in Eqns 4 and 6.

Fig S7: Response of connected cells to a sudden local osmotic shock: Influence on A) cell volume, B) the hydrostatic pressure difference ΔP and C) the osmotic pressure difference $\Delta \Pi$.”

R1.9: In Eq. 4, the cell was assumed to be a sphere with a radius of r_c . After converting the discrete model to a continuum formulation, r_c still appears in Eq 7. Have the authors considered it as a function of cell volume (V_c)? What would happen if the authors consider a different shape for the cells rather spheres?

As the reviewer correctly highlights, the cell radius r_c appears in the continuum formulation. This stems from an extension of the 2-cell volume and ion equations (Eqns 4 and 6), as the ion and fluid fluxes depend on the cellular surface area (which in turn depends on r_c). In response to the reviewer we have analyzed the opposite extreme to spherical cells, namely highly elongated cells. Our simulations suggest that with increasing cell elongation, cells become increasingly resistant to volume changes under an equivalent load. Beyond, our model may be extended for the consideration of more erratic and complex cell shapes, but such studies would require a discretely defined geometry and finite element analysis of individual cells to describe local variations in wall stretch and solute flux. Please note the following additions to the revised manuscript:

“In our main analyses we assume that the cells retain an approximately spherical shape. However, in proliferating clusters cells may attain high aspect ratios and more irregular configurations. In this section we propose a model extension to analyze highly elongated shapes for which a cylindrical geometry is a reasonable representation. In a thin-walled cylindrical vessel, the circumferential and longitudinal stress can be related to the pressure difference across the wall by $\sigma_{\theta,i} = \Delta P_i r_i / h_i$ and $\sigma_{l,i} = \Delta P_i r_i / 2h_i$, respectively. From classic Hookean relations, the change in radius and cylinder length may then be expressed as

$$\Delta r_i = \frac{r_0 \Delta P_i r_i}{K h_i} \left(1 - \frac{\nu}{2}\right), \quad \text{and}$$

$$\Delta l_i = \frac{l_0 \Delta P_i r_i}{K h_i} \left(\frac{1}{2} - \nu\right) \quad (\text{S19})$$

respectively, where l_0 is the initial cylindrical length and ν is the material Poisson’s ratio. Initially considering an arbitrary cylinder comprised of an incompressible material ($\nu = 0.5$) and of fixed reference volume V_0 , we find that with increasing reference length to radius ratio l_0/r_0 the cylinder becomes increasingly resistant to volume change under an equivalent load (Figure S4A).

We next extend our mechano-osmotic model to simulate the behavior of elongated cylindrical cells in response to an applied growth stress. Combining Eqns S18 and S19 the change in cellular radius can be expressed as $\Delta r_i = (r_0/K)(3\Delta P_i r_i/(4h_i) - \sigma_{a,i}/2)$. Assuming for illustrative purposes that the cortex is incompressible, the length change then reduces to $\Delta l_i = h_0 \sigma_{a,i}/2K$ due to the Poisson effect. For the flux relations, the cellular surface area $S_i = 2\pi l r_i + 2\pi r_i^2$ and we assume that the opening of mechanosensitive channels (via Eqn S15) depends predominantly on the circumferential stress as $\sigma_{\theta,i} = 2\sigma_{l,i}$. All other model elements and parameters remain as defined for the standard mechano-osmotic

model. Under an applied solid growth stress σ_g , a volume difference between the loaded and unloaded cell emerges as per our analysis of spherical cells. However, in keeping with the mechanisms highlighted for an arbitrary cylinder, with an increasing reference length to radius ratio l_0/r_0 the volume difference is predicted to reduce (Fig S4B). This indicates that, as cells elongate, they become less sensitive to loading. Beyond, our model may be extended for the consideration of more erratic and complex cell shapes, but such studies will require discrete geometry definitions and finite element analysis to describe local variations in wall stretch and solute flux.

Fig S4: A) Influence of cylindrical length to radius ratio l_0/r_0 on volume change from a fixed reference volume $V_0 = 1500 \mu\text{m}^3$ under constant pressure with $\Delta P/K = 0.025$; B) Difference in volume $V_2 - V_1$ of connected cylindrical cells under an applied solid growth stress $\sigma_{g,1} = 150 \text{ Pa}$.”

R1.10: Please double-check Eq.9. I guess the right-hand side should be $\Delta\Pi_{i-1} + \Delta\Pi_{i+1} - 2*\Delta\Pi_i$

We thank the reviewer for highlighting this point. The equation has been corrected in the revised manuscript. We have also rephrased the equation in terms of pressure differences for consistency with the model extension for advective flow as suggested by the other reviewers:

“For a cell within a longer series, this can be extended to develop an expression for the Laplacian of the hydrostatic pressure, whereby:

$$r_0^2 \nabla^2(\Delta P) \approx \Delta P_{i-1} + \Delta P_{i+1} - 2\Delta P_i. \quad (7)$$

We thank the reviewer for his/her supportive and constructive comments.

Reviewer #2 (Remarks to the Author):

The paper by McEvoy et al entitled Gap Junctions Amplify Spatial Variations in Cell Volume in Proliferating Solid Tumors develops a model describing the role of ion and water permeation across gap junctions to regulate cell volume in a cancer aggregate submitted to internal and external mechanical constraints.

The model implements, from first principles, the multiple sources of flux of ions and water fluxes across various channel and pump types. It considers passive transports across aquaporins, ion channels and other mechanosensitive channels of ion and water between the cell cytoplasm and the intercellular gap. Active transport is also considered. Transport across the gap junction that directly relates the cytoplasm between two cells is treated individually. The mechanical stability of the cells is also modeled. The model is first worked out in a cell doublet. It is then further turned into a continuum expression that allows the authors to predict the radially averaged behavior of cell volume in an aggregate.

The conclusions of the model are strengthened by simple but reliable experimental measurements of the role of gap junctions in maintaining a cell volume gradient in the aggregate.

Overall the paper is clearly written, the calculations look sound to the best of my knowledge, the fit with the experimental data are satisfactory. Overall I think the work is of interest. I think it could benefit from some rewriting and clarifications in certain parts.

I have some suggestions regarding the model and the experimental parts. My main suggestion is that the hypothesis of the model should be better explained in the main text or in the supplementary material.

We are grateful for these suggestions. We have now rewritten significant parts of the manuscript in order to better explain the model hypothesis. Please note the following excerpts in the revised manuscript:

“We hypothesize that gap junctions play a key role in the regulation of volume in connected cells, amplifying spatial variations in cell volume by mediating solute flow in response to mechanical loading. We therefore propose a mechano-osmotic model for the analysis of fluid and ion transport between connected cells and their environment.”

“We hypothesized that gap junctions could play a key role in the volume dynamics of connected cells, by mediating solute flow as driven by cell deformation. Volume control depends on an interplay between multiple cellular constituents, including gap junctions, mechanosensitive ion channels, energy-consuming ion pumps, and the actomyosin cortex, that coordinate to manipulate cellular osmolarity. In connected cells, our model suggests that mechanical loading significantly affects how these components cooperate to transport ions, and precise volume control is impacted by the evolution of osmotic pressure gradients between cells.”

Further, we have provided extensive discussion on the underlying thermodynamic basis for our model, as detailed below in response to the reviewer’s specific comments. Here we also briefly highlight the key improvements we have made to the revised manuscript motivated by the reviewer’s comments:

- Modification of model formulations in accordance with channel selectivity
- Extensive supplementary analysis on electro-osmotic flow in connected cells
- A detailed thermodynamic basis for the model framework
- Detailed discussion of parameter motivation
- Analysis of additional data from a cancer patient biopsy sample
- Clarifications on description of active ion pumping and cortical elasticity
- Expanded analysis of external hydrostatic and osmotic pressure gradients
- Increased emphasis on model throughout manuscript and clarification of hypothesis
- Title modification to better reflect nature of analyzed system

R2.1: In the case of passive transport across the channels (aquaporins, ion channels and gap junctions) the authors consider only the terms linked to the difference of hydrostatic and osmotic pressure. This term originates as a simplification of Onsager gradients of the chemical potentials. It does not account for advective transport. In the case of aquaporins and ion channels the assumption is common. Gap junctions have diameters that are about ten times the size of a water molecule and similar to pores created by toxins (eg hemolysin). In such cases the role of water advection and electromostic flows in ion transport of ions have been extensively addressed. Neglecting these flows is an assumption of the model, that simplifies its tractability. Since the main point of the paper is precisely the role of connexins, I guess such

hypothesis should be commented. A way to implement the advection of water in the gap junction would be to have a distinct proportionality factors relating the change in volume with osmotic pressure and hydrostatic pressure. It would amount considering a Poiseuil flow among the gap junction: eq 1=>dV/dt =-Sg ($\omega g P \Delta P - \omega g \Pi \Delta \Pi$). Such an assumption have been used in a recent paper describing lumen growth in blastocyst: Chan, C.J., Costanzo, M., Ruiz-Herrero, T. et al. Hydraulic control of mammalian embryo size and cell fate. Nature 571, 112–116 (2019). <https://doi.org/10.1038/s41586-019-1309-x>

Similarly in the context of lumen growth the fluxes across channels and gaps have been worked out in the <https://www.pnas.org/content/115/21/E4751> Dasgupta et al Physics of lumen growth.

We thank the reviewer for highlighting this valid point. In our revised manuscript we have now provided a detailed thermodynamic motivation for the model, including a discussion of Onsager gradients and the Staverman reflection coefficient. Please note the following excerpts in the revised manuscript:

“We can thus modify our dissipation function such that

$$\Phi = (\dot{n}_w \bar{V}_w + \dot{n}_s \bar{V}_s) \Delta P + \left(\frac{\dot{n}_s}{c_s^*} - \frac{\dot{n}_w}{c_w} \right) \Delta \Pi, \quad (S6)$$

whereby the osmotic pressure difference $\Delta \Pi = RT \Delta c_s$ via van't Hoffs relation. Considering this function represents a special case of the general expression $\Phi = \sum_i J_i X_i$, where J_i denotes a flow and X_i is the generalized conjugated force, the forces $X_v = \Delta P$ and $X_D = \Delta \Pi$ identify the conjugate flows $J_v = \dot{n}_w \bar{V}_w + \dot{n}_s \bar{V}_s$ and $J_D = \dot{n}_s / c_s^* - \dot{n}_w / c_w$. Following the general theory of irreversible thermodynamics and Onsagar reciprocal relations¹¹, the flows may also be expressed as:

$$J_v = L_p \Delta P + L_{pD} \Delta \Pi \quad (S7)$$

$$J_D = L_{Dp} \Delta P + L_D \Delta \Pi,$$

where the L 's are phenomenological coefficients that govern the membrane permeability and $L_{Dp} = L_{pD}$. Kedem *et al*¹² further highlighted that these coefficients could be related through the Staverman reflection coefficient¹³, o_r , leading to $L_{pD} = -o_r L_p$. This coefficient indicates the level of membrane selectivity, with $o_r = 1$ denoting an ideally selective (channels non-permeable to solutes) membrane and $o_r = 0$ a fully non-selective membrane. We note the reflection coefficient is typically denoted by σ but is instead here represented by o_r to avoid confusion with the standard notation for stress. They proposed that the flow equations can thus be rearranged such that

$$J_v = L_p (\Delta P - o_r \Delta \Pi) \quad \text{and} \quad (S8)$$

$$\dot{n}_s = c_s^* L_p (1 - o_r) \Delta P + [\omega - c_s^* L_p (1 - o_r) o_r] \Delta \Pi, \quad (S9)$$

where ω is a coefficient that relates to solute permeability.”

As correctly suggested by the reviewer, this expanded motivation for the model indicates that ion flow through gap junctions is not simply driven by hydrostatic and osmotic pressure gradients, but depends on channel selectivity. As the reviewer has indicated, gap junctions are significantly larger than water molecules and ions, and we now therefore assume they are fully non-selective. This leads to a modified form of our previous equations that includes consideration of advective flow. Please note the following additional text in the revised manuscript:

“With a diameter in the range of 1.5 – 2 nm, GJs may be approximated as fully non-selective to water molecules (diameter ≈ 0.275 nm) and ions (diameters $\approx 0.1 - 0.2$ nm) such that $o_r = 0$ ⁶. Eqns S8 and S9 then reduce to

$$J_{v,g,i} = -L_{p,g}(P_i - P_{i+1}) \quad \text{and} \quad (S10)$$

$$\dot{n}_{g,i} = -c_s^* L_{p,g}(P_i - P_{i+1}) - \omega_g(\Pi_i - \Pi_{i+1}). \quad (S11)$$

The change in cellular volume associated with fluid flow through GJs may then be written as

$$\frac{dV_{c,i}}{dt} = A_g J_{v,g,i} = -A_g L_{p,g}(P_i - P_{i+1}), \quad (S12)$$

where A_g is the surface area of the connected membrane. Similarly, the rate of change in the total number of ions in a cell is then given by

$$\frac{dN_i}{dt} = A_g \dot{n}_{g,i} = -A_g \left(c_s^* L_{p,g}(P_i - P_{i+1}) + \omega_g(\Pi_i - \Pi_{i+1}) \right), \quad (S13)$$

where the first bracketed term accounts for advection ion flow and the second term describes diffusive transport.”

Further, we have also provided a detailed supplementary analysis of electro-osmotic flow and multiple ion species in SI Section S8. Our simulations suggest that our reduced mechano-osmotic model captures the key trends predicted by this more detailed analysis. Please note the following excerpts from the revised manuscript:

“Simulations reveal that the MEO [mechano-electro-osmotic] model predicts the same trends as the mechano-osmotic model discussed in the main paper text (Fig 2): the volume of a loaded cell reduces with increasing applied stress and its neighbor increasingly swells due to an ion influx via gap junctions (Fig S6A).”

We have also now referenced the works of Chan *et al.* and Dasgupta *et al.* in the revised manuscript:

“Beyond, similar models have been proposed to understand how fluid or ion transport governs the swelling or shrinkage of adhered single cells^{14,15} and lumen growth^{16,17}”.

R2.2: The assumption that the passive cortical stress is proportional to the difference of the surface to a reference surface is also a very idealized though practical assumption, assuming that cell cortex is equivalent to an elastic material. It has been used in various forms in vertex models. Here the assumption of a S_0 , to some extent, is equivalent to the assumption of a preferred volume. Cortices relax within 10 of seconds and the passive stress vanishes as fast. The assumption of an elastic shell is not uncommon but often misleading. I would suggest that the authors refer to this caveat.

Viewing the cortex as a viscoelastic material, it may be considered to have an effective initial and long-term elastic modulus whereby although it will exhibit stress relaxation the stress will not entirely dissipate. As the timescales associated with volume equilibration (10s of minutes) are larger than cortical dissipation (10s of seconds), we can therefore neglect the influence of material viscosity and view the effective stiffness as a long-term modulus. Please note the following text in the revised manuscript:

“As per recent cortex models^{18–20}, we begin with the consideration of a general viscoelastic model such that the passive stress may be expressed by $\sigma_{p,i} = K(A_i/A_0 - 1)/2 + \eta(1/A_i)(dA_i/dt)$, where K is the effective cortical stiffness, η is the effective viscosity of the cell cortex, and A_0 is a reference surface area. The apparent cortex viscosity has been reported to lie in the range of $10^2 - 10^3 \text{ Pa}\cdot\text{s}$ ^{21,22}, and under loading the cell radius has been reported to change by ~10% in several minutes²³. As such, the viscous stress $\eta(2/r)(dr/dt) \approx 0.1 - 1 \text{ Pa}$ is much smaller than the elastic terms in the passive stress

expression. Therefore, we consider the contribution of the viscous term to be negligible in this analysis with a view that K denotes the long-term cortical stiffness, and the total stress reduces to $\sigma_i = (K/2)(r_{c,i}^2/r_o^2 - 1) + \sigma_{a,i}$ for a spherical cell of radius $r_{c,i}$.”

Further we agree with the reviewer that assuming the cortex as elastic, or even viscoelastic, is an idealization. Realistically the cytoskeleton actively remodels, with myosin motors actively generating force as dependent on stress-mediated signaling pathways. We have extensively considered such interactions and feedback in previous work^{24,25}, but within the current study these additional considerations would add a level of complexity that would distract from the key mechanisms of multicellular volume regulation, and we therefore simply assume a constant active stress in the current framework. We intend to explore the evolution of active forces and signaling feedback with ion transport in future studies. Please note the following text the revised manuscript:

“An investigation more directed to the specific influence of cortical organization and myosin contractility could readily extend this expression to describe long-term remodeling in response to signaling and stress in accordance with our previous work^{24,25}.”

“Future advancements should also focus on detailing the interdependence between dynamic actomyosin contractility and cell osmolarity, building on our previous work to further understand how the two-way feedback between stress and signaling guides nuclear gene expression²⁶, dynamic force generation²⁷, and cancer invasion²⁸.”

R2.3: I appreciated the table in supplementary material of the paper summarizing the numerical values that are used in the text. It refers to the authors’ own measurements and to that from Jiang and Sun. It would be more informative to explain in supplementary how the values were measured in these two papers and to give a range of what the authors feel “acceptable”. Which of these parameters were used to fit of the experimental data. Where does the 63.3 value comes from? Why 10% of the reference area? Is the 3.3 really relevant?

We have now provided motivation for all parameters following the table. Additionally, phase maps for key parameters are now also provided to visualize their influence on cell volume (Fig S3). As highlighted by the reviewer we chose the adhered membrane area as 10% of the reference area. In lieu of clear evidence for contact area of cell-cell adhesions, 10% is a reasonable approximation for cells contacted on all sides within clusters and also permits the GJ permeability to lie within the derived range. Please note the following text in the revised manuscript:

“The density of GJs on the adhered membrane of epithelial-like cells has been estimated⁴ to lie in the range of $\rho_g \approx 2 \times 10^{14} m^{-2}$. Assuming Poiseuille flow through GJs of radius $r_g \approx 1 nm$ and length $l_g \approx 15 nm$ ⁵, following Mathias *et al* (2008)⁴ and Gao *et al.* (2011)⁶, the solute permeability factor can be estimated as $L_{p,g} = \rho_g \pi r_g / (8 \eta l_g)$, where η is the viscosity of water. Within channels of radius $1 nm$ and at body temperature ($T = 310 K$), the viscosity of water can be approximated as $0.5 mPa \cdot s$ ⁷, leading to a coefficient $L_{p,g} \approx 10^{-11} m \cdot s^{-1} \cdot Pa^{-1}$. Further, considering the molar volume of water $\bar{V}_w = 18 mL/mol$, we can determine the solvent GJ permeability coefficient in terms of the number of moles whereby $\omega_{g,w} = L_{p,g} / \bar{V}_w = 5.82 \times 10^{-7} mol \cdot m^{-2} s^{-1} Pa^{-1}$. Assuming similar diffusive behavior for the solutes (ions), $\omega_{g,w}$ suggests that ion coefficient ω_g should lie in the range $10^{-7} - 10^{-6}$. We determined that a value of $\omega_g = 5 \times 10^{-7} mol \cdot m^{-2} s^{-1} Pa^{-1}$ provides good agreement between our simulated and experimentally measured volumes, assuming a cell-cell adhered membrane surface area A_g on the order of 10% of the reference cell surface. Across a number of cell types the membrane solvent permeability rate P_s has been reported to lie in the range of $10^{-4} - 10^{-5} m/s$ ²⁹. Jiang and Sun (2013)¹⁹ highlighted that this can be related to the membrane water permeability factor

such that $L_{p,m} = P_s \bar{V}_w / RT$, from which we ascertain an upper value of $L_{p,m} = 7 \times 10^{-12}$ used here. The spheroid radius r_{max} is directly measured from the day 5 experimental images (Fig 3). In epithelial cells, the thickness h of the actin cortex has been reported to vary from $0.1 \sim 0.6 \mu m$ ³⁰ for which we assume the upper value. Experimentally reported values for cell stiffness are highly variable ($0.1 - 100 kPa$ ³¹), likely associated with measurement timescales and cell remodeling. We assume a value $K = 6 kPa$ and $\sigma_a = 100 Pa$ in line with Jiang and Sun (2013), with treatment of dissipative effects discussed in Section S4. The reference cell radius r_0 was approximated such that the predicted cell volumes at an early stage of organoid growth provide good agreement with our experiments (Fig S10E). The external osmotic pressure Π^{ext} can be computed from physiological ion concentrations ($c_{Na^+,e} = 145mM$, $c_{K^+,e} = 5mM$ and $c_{Cl^-,e} = 110mM$ ³²) with $\Pi^{ext} = RT(c_{Na^+,e} + c_{K^+,e} + c_{Cl^-,e}) = 0.67 MPa$. As the free energy from ATP is $\Delta G_a \approx 30 kJ/mol$, the critical osmotic pressure for active pumping is then $\Delta \Pi_c = \Pi^{ext} \Delta G_a / RT = 40 GPa$. The flux associated with active ion pumps has been measured to lie between 10^{-7} and $10^{-6} mol.m^2.s^{-1}$ ^{33,34}. As per Jiang and Sun (2013), after dividing by $\Delta \Pi_c$ the pump coefficient should be confined to the range $\gamma \approx 2.5 \times 10^{-18} \sim 2.5 \times 10^{-17} mol.m^{-2}.s^{-1}.Pa^{-1}$, for which we choose a value of $2.25 \times 10^{-17} mol.m^{-2}.s^{-1}.Pa^{-1}$. The ion fluxes across mechanosensitive and leak channels have an amplitude on the order of active ion pumps (i.e. $10^{-7} \sim 10^{-6} mol.m^2.s^{-1}$)^{33,34}; To maintain such continuity Jiang and Sun (2013) suggested a value $\beta = 2 \times 10^{-11} mol.m^{-2}.s^{-1}.Pa^{-2}$, which we retain for our analyses, while the threshold and saturation stress are reduced to provide better agreement with our experiments (Fig 3). In keeping with this motivation, the leak channel permeability factor ω_l is assumed to equal the threshold permeability of the MS channels (i.e. $\beta \sigma_c$). The influence of key model parameters on cell volume in two connected cells (in response to solid growth stress $\sigma_{g,1}$ on cell 1) is shown in Fig S3.”

R2.4: Before figure 1, the sentence : We consider there to be a critical... energy input is not really clear to me. If I understand correctly you assume that the active flux of pumps is not constant but depends on the difference of the osmotic pressure to a critical value $\Delta \Pi_c$. Are there experimental evidences of this, rather than a constant active flux. If yes it may be worth mentioning.

We agree with the reviewer that the mechanism was not clearly expressed in the initial text. Active ion pumps will transport ions to/from the cytosol as long as the process is energetically favorable ($\Delta G < 0$); It cannot require more energy to move an ion against the concentration gradient than is provided by ATP hydrolysis. If the osmotic pressure difference exceeds a critical value (determined by $\Delta G = 0$) then it becomes increasingly likely that the pump will change direction (reversal potential)³⁵. Please note the following text in the revised manuscript:

“These ion pumps require an energy input, such as from ATP hydrolysis, to overcome the energetic barrier associated with moving ions against the concentration gradient. Following Jiang and Sun (2013)¹⁹, the free energy change associated with pumping action can be expressed as $\Delta G = RT \log (c_i / c_{ext}) - \Delta G_a$, where ΔG_a is an energy input is associated with hydrolysis of ATP. The ion flux associated with active pumping can then be written as $\dot{n}_{p,i} = \gamma' \Delta G$, where γ' is a permeation constant. Maintaining our dilute assumption, ΔG can be linearized as $\Delta G = RT (\Pi_i - \Pi^{ext}) / \Pi^{ext} - \Delta G_a$. We can therefore identify a critical osmotic pressure difference $\Delta \Pi_c$, determined when $\Delta G = 0$, such that $\Delta \Pi_c = \Pi^{ext} \Delta G_a / RT$ (noting that when $\Delta G > 0$ active pumping is no longer energetically favorable and the pumping direction will reverse³⁵). Thus the ion flux generated by active pumping by ion transporters can be expressed as $\dot{n}_{p,i} = \gamma (\Delta \Pi_c - \Delta \Pi_i)$, where γ is a rate constant.”

We further expand on this point within our consideration of electro-osmotic flow and multiple ion species. Please not the following additions to the revised manuscript:

“From Glitsch and Tappe (1995)³⁵, the free energy during the pumping action is $\Delta G = \Delta G_a + 3 \left(-F\phi_i + RT \ln \left(\frac{c_{Na^+,e}}{c_{Na^+,i}} \right) \right) + 2 \left(F\phi_i + RT \ln \left(\frac{c_{K^+,e}}{c_{K^+,i}} \right) \right)$, where ΔG_a is an energy input associated with hydrolysis of ATP. Provided this reaction is energetically favorable ($\Delta G < 0$), the pump transfers three Na^+ ions against their concentration gradient to the cytosol, and two K^+ from the external media into the cytosol. However, the net result of this complex pathway, which downstream is predicted to cause an overall increase in cytosolic osmotic pressure and cell volume (Fig S5H-I), is similar to the behavior captured by only considering an active influx in Eqn 6.”

R2.5: It would be good to indicate in the text, how many spheroids you used to compare the model to the experiments. In figure 3 and 4, each point corresponds to one distance from the center but I guess they originate from some experimental replica.

We thank the reviewer for highlighting this omission. Five to fifteen multi-cellular clusters were measured in each experiment, now indicated within the experimental methodology and relevant figure captions. Please note the following additional text:

“Five to fifteen multi-cellular clusters were measured in each experiment”.

R2.6: The first paragraphs of the discussion until :”gap junctions play a critical...” summarizes the scenario that your model display based on the assumptions you made. It reads very much though as conclusions from experimental data that would probe the mechanism one after the other. I would rather prefer that the author clearly state that the scenario originating from the model that couple these many fluxes is compatible with experimental results showing gradient of cell volume in spheroids and there inhibition when gap junctions are altered. I suggest putting more emphasis on the fact that the model is an interpretation based on the assumptions. I am not a great fan of the last two sentences but it is there a matter of taste.

We thank the reviewer for these suggestions. Within the initial discussion paragraphs we have now rephrased to highlight that our conclusions are drawn from modeling insight, based on interpretation of the experiments. Please note the following modified excerpts from the revised manuscript:

“In connected cells, our model suggests that mechanical loading significantly affects how these components cooperate to transport ions, and precise volume control is impacted by the evolution of osmotic pressure gradients between cells.”

“This proposed mechanism is supported by the blocking of gap junctions, which is revealed to prevent such volume changes and, also, reduce shrinkage of the loaded cell.”

“Our model indicates that this can relieve cellular membrane tension, leading to the closure of mechanosensitive ion channels; A reduction in this channel permeability would dictate that fewer ions are lost to the interstitium.”

“With a radial reduction in solid stress loading, our simulations suggest that more ion channels remain open towards the organoid periphery, allowing peripheral cells to maintain lower osmotic pressures. Following the mechanisms highlighted by our two-cell analysis, this could drive a radial intercellular flow of ions from cells in the core, thereby increasing the ion concentration in cells situated towards the periphery.”

We also note the inclusion of additional data from a breast cancer biopsy to highlight that our model predictions are relevant for in-vivo cell behavior. Please note the following additions to the revised manuscript:

“Further, in recent work³ we obtained grade2 ER+ invasive ductal carcinoma breast cancer samples from a human patient, which were fixed, sectioned and stained for imaging with confocal microscopy (Fig S11A). Within the tumor mass, spheroidal acinar clusters of cells surrounded by basement membrane were identified, which share characteristics with the 3D experimental model data reported in this study. In these acinar structures, nuclear volumes were observed to increase from core to periphery, consistent with our simulations (Fig S11B).”

We have additionally removed the generalization from the final two sentences, suggested the mechanisms should be investigated in other cancer cell lines, and softened the statement regarding use of our model for therapeutic design:

“In summary, our findings suggest that intercellular ion flow may be an important mediator of breast cancer progression. Future studies should aim to characterize this behavior across alternative cancer cell lines to assess if these results are generalizable to other types of cancer. Our proposed model could also be extended to aid the development of therapeutics that target inter- and extra-cellular ion transport.”

R2.7: Along those lines, the title should reflect more the nature of the experimental system. MCF10 aggregates are far from being perfect in vitro replica of tumors. It would be fairer in to mention say ” Tumorigenic cell aggregates” or “ tumor spheroids” rather than tumors.

We appreciate the reviewer’s suggestion and have updated the title as follows:

“Gap Junctions Amplify Spatial Variations in Cell Volume in Proliferating Tumor Spheroids ”.

All in all I believe the article is sound and consistent with its assumption. It describes fairly the experimental data and is of interest to the readers of Nature Com.

We again sincerely thank the reviewer for this supportive statement, and his/her fair and well-thought-out comments, which have helped us immensely.

Reviewer #3 (Remarks to the Author):

In this manuscript entitled ""Gap Junctions Amplify Spatial Variations in Cell Volume in Proliferating Solid Tumors", McEvoy et al. consider as a building block a model of individual cell volume regulation (Ref. 26) in a multi-cellular context (first two cells and then a cell spheroid) by adding gap junctions that enable solute and solvent flows between neighbouring cells. They use the model to explain 1) how the local application of a solid stress to one cell in a doublet can lead to a differential shrinking and swelling 2) their experimental observation previously published (Ref 11) that cells in a spheroid core have a smaller nuclear volume than cell in the spheroid periphery.

The manuscript is well written and clear and the topic interesting and new but I cannot recommend its publication given some important conceptual problems in the model that I could not understand and which I detail below.

Thank you very much for the detailed comments. We have addressed each comment carefully and have extensively revised the model following your valuable suggestions. In particular, we would like to highlight the following key improvements we have made to the revised manuscript as motivated by the reviewer’s comments:

- A detailed thermodynamic basis for the model framework

- Modification of model formulations in accordance with channel selectivity
- Extensive supplementary analysis on electro-osmotic flow in connected cells
- Detailed discussion of parameter motivation
- Expanded analysis of external hydrostatic and osmotic pressure gradients
- Clarifications on description of active ion pumping and cortical elasticity

R3.1: I do not understand equation (1) of the paper. Why would this model of fluid transport hold for gap junctions ? Any soluble molecule smaller than 2 nm (i.e. ions such as Na, K, Cl) can go through a gap junction which is therefore not selective. In the absence of this selectivity assumption, how can the authors justify this formula (Kedem and Katchalsky, J Gen Physiol (1961) 45 (1): 143–179., Staverman, Trans. Faraday SOC., 1952, 48, 176.) ? Why isn't it only the difference of osmotic pressures between the non-permeable osmolites that enters the thermodynamic driving force ? Same question in equation (4), it seems to me that ion motion through selective and non-selective channels should involve different thermodynamic forces. In my opinion, the authors should clarify the thermodynamic framework they use and also give references.

We sincerely thank the reviewer for highlighting these points; We believe that thoroughly addressing these concerns has significantly improved the quality of our manuscript. We have greatly expanded our description of the thermodynamic framework, including a discussion of channel selectivity. Please note the following additions to the revised manuscript:

“Here we derive equations of solute and solvent flow between connected solutions, as per the work of Kedem *et al.* (1958)¹². As highlighted by Kedem *et al.*, entropy production is the starting point of any thermodynamic description of non-equilibrium systems. Initially considering a 2-cell system in which solutions are separated by a membrane, entropy production $d_i S/dt$ is given by

$$\frac{d_i S}{dt} = \frac{1}{T} (\mu_w^{i+1} - \mu_w^i) \frac{dN_w^i}{dt} + \frac{1}{T} (\mu_s^{i+1} - \mu_s^i) \frac{dN_s^i}{dt}, \quad (S1)$$

where T is the absolute temperature, μ_w (μ_s) denotes the chemical potential of the solvent (solute), and dN/dt is associated number of moles passing into cell i per unit time. The dissipation (per unit area) may then be written as

$$\Phi = \frac{T}{A} \frac{d_i S}{dt} = (\mu_w^{i+1} - \mu_w^i) \dot{n}_w + (\mu_s^{i+1} - \mu_s^i) \dot{n}_s, \quad (S2)$$

where $\dot{n}_w = (1/A) dN_w^i/dt$ and $\dot{n}_s = (1/A) dN_s^i/dt$. With an approximation that the chemical potentials for ideal solutions are appropriate, the chemical potential difference is given by

$$\mu^{i+1} - \mu^i = \bar{V} \Delta P + RT \Delta \ln(\gamma), \quad (S3)$$

such that \bar{V} is the partial molar volume, ΔP is the hydrostatic pressure difference between the cells, γ is the molar fraction of the constituent, and R is the gas constant. In a dilute solution, where the volume fraction of the solute is relatively small (i.e. $c_s \bar{V}_s \ll 1$), Eqn S3 can be rewritten as

$$\mu_s^{i+1} - \mu_s^i = \bar{V}_s \Delta P + RT \Delta \ln(c_s) = \bar{V}_s \Delta P + RT \frac{\Delta c_s}{c_s^*}, \quad (S4)$$

where c_s^* is the mean solute concentration across both cells and $\Delta c_s/c_s^* = \ln(c_s^{i+1}/c_s^i)$ if $\Delta c_s/c_s^i \ll 1$. Similarly, the chemical potential difference for the solvent is

$$\mu_w^{i+1} - \mu_w^i = \bar{V}_s \Delta P + RT \frac{\Delta c_s}{c_w}. \quad (S5)$$

where to good approximation, $c_w = 1/\bar{V}_w$. We can thus modify our dissipation function such that

$$\Phi = (\dot{n}_w \bar{V}_w + \dot{n}_s \bar{V}_s) \Delta P + \left(\frac{\dot{n}_s}{c_s^*} - \frac{\dot{n}_w}{c_w} \right) \Delta \Pi, \quad (S6)$$

whereby the osmotic pressure difference $\Delta \Pi = RT \Delta c_s$ via van't Hoffs relation. Considering this function represents a special case of the general expression $\Phi = \sum_i J_i X_i$, where J_i denotes a flow and X_i is the generalized conjugated force, the forces $X_v = \Delta P$ and $X_D = \Delta \Pi$ identify the conjugate flows $J_v = \dot{n}_w \bar{V}_w + \dot{n}_s \bar{V}_s$ and $J_D = \dot{n}_s / c_s^* - \dot{n}_w / c_w$. Following the general theory of irreversible thermodynamics and Onsagar reciprocal relations¹¹, the flows may also be expressed as:

$$J_v = L_p \Delta P + L_{pD} \Delta \Pi \quad (S7)$$

$$J_D = L_{Dp} \Delta P + L_D \Delta \Pi,$$

where the L 's are phenomenological coefficients that govern the membrane permeability and $L_{Dp} = L_{pD}$. Kedem *et al*¹² further highlighted that these coefficients could be related through the Staverman reflection coefficient¹³, o_r , leading to $L_{pD} = -o_r L_p$. This coefficient indicates the level of membrane selectivity, with $o_r = 1$ denoting an ideally selective (channels non-permeable to solutes) membrane and $o_r = 0$ a fully non-selective membrane. We note the reflection coefficient is typically denoted by σ but is instead here represented by o_r to avoid confusion with the standard notation for stress. They proposed that the flow equations can thus be rearranged such that

$$J_v = L_p (\Delta P - o_r \Delta \Pi) \quad \text{and} \quad (S8)$$

$$\dot{n}_s = c_s^* L_p (1 - o_r) \Delta P + [\omega - c_s^* L_p (1 - o_r) o_r] \Delta \Pi, \quad (S9)$$

where ω is a coefficient that relates to solute permeability.”

Building on this thermodynamics description, we have modified our model as suggested by the reviewer to explicitly consider the non-selectivity of gap junctions. Please note the following in the revised text:

“With a diameter in the range of 1.5 – 2 nm, GJs may be approximated as fully non-selective to water molecules (diameter ≈ 0.275 nm) and ions (diameters $\approx 0.1 - 0.2$ nm) such that $o_r = 0$ ⁶. Eqns S8 and S9 then reduce to

$$J_{v,g,i} = -L_{p,g} (P_i - P_{i+1}) \quad \text{and} \quad (S10)$$

$$\dot{n}_{g,i} = -c_s^* L_{p,g} (P_i - P_{i+1}) - \omega_g (\Pi_i - \Pi_{i+1}). \quad (S11)$$

The change in cellular volume associated with fluid flow through GJs may then be written as

$$\frac{dV_{c,i}}{dt} = A_g J_{v,g,i} = -A_g L_{p,g} (P_i - P_{i+1}), \quad (S12)$$

where A_g is the surface area of the connected membrane. Similarly the rate of change in the total number of ions in a cell is then given by

$$\frac{dN_i}{dt} = A_g \dot{n}_{g,i} = -A_g \left(c_s^* L_{p,g} (P_i - P_{i+1}) + \omega_g (\Pi_i - \Pi_{i+1}) \right), \quad (S13)$$

where the first bracketed term accounts for advection ion flow and the second term describes diffusive transport.”

R3.2: Again equation (2) is unclear to me. Why is it that the total osmotic pressure comes as a generalized thermodynamic force and not from the chemical potential of

each ion specie ? Also each ion specie may have its own mobility through the gap junctions which may in particular depend of the ion charge. The authors do introduce a work of caution about this in the conclusion but this is not enough to have full trust in the model in my opinion. Again the full thermodynamic framework at the origin of these kinetic relations needs to be clarified.

We have partly addressed this concern in response to the reviewer’s previous comment (R3.1). Equation 2 has now been updated to reflect the influence of channel non-selectivity as motivated by the detailed thermodynamic description in Section S1-S2. Please note the following additional text in the revised manuscript:

“The advective/diffusive behavior may then be characterized by an ion flow given by $\dot{n}_{g,i} = -c_s^* L_{p,g}(P_i - P_{i+1}) - \omega_g(\Pi_i - \Pi_{i+1})$, where ω_g is rate constant and c_s^* is the mean solute concentration across the two connected cells. Under these conditions the rate of change in the total number of ions in a cell can be determined:

$$\frac{dN_i}{dt} = -A_g \left(c_s^* L_{p,g}(P_i - P_{i+1}) + \omega_g(\Pi_i - \Pi_{i+1}) \right). \quad (2)$$

Further, we have now provided a detailed supplementary analysis on the influence of multiple ion species, electroneutrality, and electrical potentials on cell volume (Section S8). *Ultimately*, we demonstrate that this analysis identifies the same key trends as our reduced mechano-osmotic model; Namely that increasing external loading on a cell drives ions across gap junctions into a connected neighbor, which consequentially swells. Please note the following additions to the revised text:

Consideration of electrical potentials

Up until this point we have confined our consideration of cellular volume regulation to a dependence on osmotic and hydrostatic pressures, and neglected the role of electrical potentials. In this section, we proceed to define a framework for coupled mechano-electro-osmotic (MEO) interactions and demonstrate that the mechanisms proposed for multi-cell volume regulation are largely unchanged by such an extension. We begin by introducing the classic pump-leak model^{36,37} to describe the fluxes of dominant ions associated with cellular volume control (Na^+ , K^+ , and Cl^-), whereby

$$\frac{dc_{Na^+,i}}{dt} = -\frac{A_i g_{Na}}{FV_i} \left(\phi_i - \frac{RT}{F} \ln \left(\frac{c_{Na^+,e}}{c_{Na^+,i}} \right) \right) - \frac{3pA_i}{FV_i}, \quad (S20)$$

$$\frac{dc_{K^+,i}}{dt} = -\frac{A_i g_K}{FV_i} \left(\phi_i - \frac{RT}{F} \ln \left(\frac{c_{K^+,e}}{c_{K^+,i}} \right) \right) + \frac{2pA_i}{FV_i}, \quad (S21)$$

$$\frac{dc_{Cl^-,i}}{dt} = \frac{A_i g_{Cl}}{FV_i} \left(\phi_i + \frac{RT}{F} \ln \left(\frac{c_{Cl^-,e}}{c_{Cl^-,i}} \right) \right). \quad (S22)$$

Here, solute concentrations $c_{j,i} = N_{j,i}/V_i$, ϕ_i is the membrane potential, A_i is the cell surface area, F is the Faraday constant, and g_j are the ion channel conductances for a given species. The log-based term is a form of the Nernst equation, determining the electrical potential of the species as a function of internal and external (e) concentrations. In this formulation, p is the strength of the pump current with associated values representing the 3:2 stoichiometry of the $Na - K$ ATPase. While earlier we assumed that pumping actively transports a single ion species across the membrane, the biological mechanism of $Na - K$ pumping is more complex. From Glitsch and Tappe (1995)³⁵, the free energy during the pumping action is $\Delta G = \Delta G_a + 3 \left(-F\phi_i + RT \ln \left(\frac{c_{Na^+,e}}{c_{Na^+,i}} \right) \right) + 2 \left(F\phi_i + RT \ln \left(\frac{c_{K^+,e}}{c_{K^+,i}} \right) \right)$, where ΔG_a is an energy input associated with hydrolysis of ATP. Provided this reaction is energetically favorable ($\Delta G < 0$), the pump transfers three Na^+ ions against their concentration gradient to the cytosol, and

two K^+ from the external media into the cytosol. However, the net result of this complex pathway, which downstream is predicted to cause an overall increase in cytosolic osmotic pressure and cell volume (Fig S5H-I), is similar to the behavior captured by only considering an active influx in Eqn 6.

The balance laws must be additionally supplemented by the electroneutrality condition

$$c_{Na^+,i} + c_{K^+,i} - c_{Cl^-,i} + z_x X/V_i = 0, \quad (S23)$$

where X is the total amount of impermeable molecules in the cell³⁷ which have a mean valence z_x . Initially considering a single suspended cell, from **Eqn S8** the volume may thus be determined from $dV_i/dt = -L_{p,m}(\Delta P_i - \Delta \Pi_i)$, where the osmotic pressure difference $\Delta \Pi_i = RT(c_{Na^+,i} + c_{K^+,i} + c_{Cl^-,i} + X/V_i - (c_{Na^+,e} + c_{K^+,e} + c_{Cl^-,e}))$ and recall that the hydrostatic pressure difference $\Delta P_i = 2\sigma_i h/r_i + \sigma_{g,i}$.

Predictions for single cell behavior

Previously we have discussed mechanosensitive (MS) channels, proteins in the cell membrane that open under a tensile membrane stress to allow solute flow as driven by their electro-osmotic potential. One of the most widely appreciated MS pathways relates to Piezo1, which opens under tension to allow a calcium influx, thereby activating Ca^{2+} -gated K^+ channels to relieve osmotic pressure in swollen cells³⁸. This behavior may be described by adapting our MS channel model (Eqn S15) to incorporate a stress-dependence in the conductivity of K^+ channels, such that

$$g_K(\sigma_i) = \begin{cases} g_{K0} & \text{if } \sigma_i \leq \sigma_c \\ g_{K0} + \beta_K(\sigma_i - \sigma_c) & \text{if } \sigma_c < \sigma_i < \sigma_s, \\ g_{K0} + \beta_K(\sigma_s - \sigma_c) & \text{if } \sigma_i \geq \sigma_s \end{cases}, \quad (S24)$$

where g_{K0} is the conductivity of leak (always permeable) channels, and β_K , σ_c , and σ_s are MS constants. Finally, Mori (2011)³⁷ demonstrated that the membrane potential may be approximated by

$$\phi_i = \frac{FV_i}{C_m A_i} \left(c_{Na^+,i} + c_{K^+,i} - c_{Cl^-,i} + z_x \frac{X}{V_i} \right), \quad (S25)$$

where C_m is the membrane capacitance per unit area. This system of equations can be solved to predict the time-dependent evolution of cell volume, membrane potential, and solute concentrations using in-built ode solvers in Matlab (*ode23s*). We assume the external ion concentrations remain at physiological levels, such that $c_{Na^+,e} = 145mM$, $c_{K^+,e} = 5mM$ and $c_{Cl^+,e} = 110mM$ ³² and that the conductivity coefficients are confined to previously suggested ranges³⁹, with $C_m = 0.01 F/m^2$, $g_{Na} = 0.1 S/m^2$, $g_{Cl} = 2 S/m^2$, and $g_K^{max} = 1 S/m^2$ such that $g_{K0} + \beta_K(\sigma_s - \sigma_c) = g_K^{max}$ and $g_{K0} = 0.1 S/m^3$. The pump current $p = 5 mA/m^2$, number of impermeable solutes $X = 2 X 10^{-13} mol$, and mean valence $z_x = -1.5$ are estimated to provide a reasonable prediction of cytosolic ion concentrations (Fig S5)³². The remaining model parameters are fixed at those previously outlined (Table S1). Following the application of stress σ_g , the cell is predicted to reduce in volume (Fig S5A), with an associated steady state increase in membrane potential due to an increase in K^+ ions (Fig S5C). The MEO equations may also be rearranged at steady state by setting the time derivatives to zero (i.e. $d/dt = 0$), reducing to the following:

$$Na_e e^{-\left(\frac{F(\phi + \frac{3p}{g_{Na}})}{RT}\right)} + K_e e^{-\left(\frac{F(\phi - \frac{2p}{g_K})}{RT}\right)} - Cl_e e^{\left(\frac{F\phi}{RT}\right)} + z_x \frac{X}{V_i} = 0, \quad (S26)$$

$$RT \left(Na_e e^{-\left(\frac{F(\phi + \frac{3p}{g_{Na}})}{RT}\right)} + K_e e^{-\left(\frac{F(\phi - \frac{2p}{g_K})}{RT}\right)} + Cl_e e^{\left(\frac{F\phi}{RT}\right)} + \frac{X}{V_i} - (Na_e + K_e + Cl_e) \right) - \Delta P_i = 0, (S27)$$

where the first expression stems from the electroneutrality condition and the second from steady state hydrostatic and osmotic pressure balance. We solve this system using the Matlab in-built function *fsolve* (Fig S5D-F) in conjunction with the mechanical constraints. This approach no longer requires the approximation for the membrane potential (Eqn S25), and shows good agreement with the transient solution (blue markers in Figure S5D-F). It is interesting to note that with consideration of electrolyte flow, the magnitude of applied stress required to induce a comparable reduction in cell volume is significantly higher than in the case non-electrolyte flow. In non-electrolyte flow, an applied stress increases cytosolic hydrostatic pressure, causing a solvent efflux. In response to a reduced fluid content the cytosolic osmotic pressure increases and thus there is also an increased solute flow to the external media. In turn, additional fluid is lost due to a depleted cytosolic ion concentration. However, with inclusion of the role of fixed charges and electroneutrality, osmotic gradients alone are not sufficient to cause an equivalent solute flow, as the electrical potential within the cytosol resists ion loss. As such the osmotic pressure remains high under single cell loading, permitting the cell to retain fluid and resist shrinkage.

Fig S5: (A-C) Single cell behavior in response to applied loading with consideration of electro-osmotic flow using Matlab *ode23s* to compute the time-series solution: predictions for A) cell volume, B) membrane potential, and C) concentrations of Na^+ , K^+ , and Cl^- over time. (D-F) Steady state single cell behavior in response to a series of applied loading with consideration of electro-osmotic flow using Matlab *fsolve*: predictions for D) cell volume, E) membrane potential, and F) concentrations of Na^+ , K^+ , and Cl^- . Markers show corresponding predictions at steady state from the transient analysis. (H-I) Influence of pump current p on H) cell volume and I) membrane potential.

Analysis of connected cells

We next proceed to extend the MEO model for the analysis of connected cells and solvent/solute flow through gap junctions (GJs). As described in Section S2, GJs are not selective for individual ion species or solvent. Combining Eqns S11 and S20, transport of a given (positively charged) ion species j through GJs (from cell $i + 1$ to cell i) may then be expressed by

$$\frac{dc_{g,j}}{dt} = -\frac{A_g g_g}{FV_i} \left((\phi_i - \phi_{i+1}) - \frac{RT}{F} \ln \left(\frac{c_{j,i+1}}{c_{j,i}} \right) \right) - \frac{A_g L_{p,g} c_j^*}{V_i} (\Delta P_i - \Delta P_{i+1}), \quad (S28)$$

where the term on the left describes diffusive flow driven by electro-osmotic gradients and the term on the right describes advective flow as driven by hydrostatic pressure gradients (as described in Section S1). Here g_g is a coefficient associated with GJ conductivity, assumed for demonstrative purposes to equal g_K^{max} , and c_j^* is the mean cytosolic concentration of ion species j across the connected cells of interest. Eqns S20-S21 can then be extended to describe the change in ion concentrations within a given cell i as dependent on exchange across both the cell membrane and GJs, whereby:

$$\begin{aligned} \frac{dc_{Na^+,i}}{dt} &= -\frac{A_i g_{Na}}{FV_i} \left(\phi_i - \frac{RT}{F} \ln \left(\frac{c_{Na^+,e}}{c_{Na^+,i}} \right) \right) - \frac{3pA_i}{FV_i} \\ &- \frac{A_g g_g}{FV_i} \left((\phi_i - \phi_{i+1}) - \frac{RT}{F} \ln \left(\frac{c_{Na^+,i+1}}{c_{Na^+,i}} \right) \right) - \frac{A_g L_{p,g} c_{Na^+}^*}{V_i} (\Delta P_i - \Delta P_{i+1}), \end{aligned} \quad (S29)$$

$$\begin{aligned} \frac{dc_{K^+,i}}{dt} &= -\frac{A_i g_K}{FV_i} \left(\phi_i - \frac{RT}{F} \ln \left(\frac{c_{K^+,e}}{c_{K^+,i}} \right) \right) + \frac{2pA_i}{FV_i} \\ &- \frac{A_g g_g}{FV_i} \left((\phi_i - \phi_{i+1}) - \frac{RT}{F} \ln \left(\frac{c_{K^+,i+1}}{c_{K^+,i}} \right) \right) - \frac{A_g L_{p,g} c_{K^+}^*}{V_i} (\Delta P_i - \Delta P_{i+1}), \end{aligned} \quad (S30)$$

$$\begin{aligned} \frac{dc_{Cl^-,i}}{dt} &= \frac{A_i g_{Cl}}{FV_i} \left(\phi_i + \frac{RT}{F} \ln \left(\frac{c_{Cl^-,e}}{c_{Cl^-,i}} \right) \right) \\ &+ \frac{A_g g_g}{FV_i} \left((\phi_i - \phi_{i+1}) + \frac{RT}{F} \ln \left(\frac{c_{Cl^-,i+1}}{c_{Cl^-,i}} \right) \right) - \frac{A_g L_{p,g} c_{Cl^-}^*}{V_i} (\Delta P_i - \Delta P_{i+1}). \end{aligned} \quad (S31)$$

The associated cell volume depends on both hydrostatic and osmotic pressure gradients, as before, with:

$$\begin{aligned} \frac{dV_i}{dt} &= -A_g L_{p,g} (\Delta P_i - \Delta P_{i+1}) \\ &- A_i L_{p,m} \left(\Delta P_i - RT \left(c_{Na^+,i} + c_{K^+,i} + c_{Cl^-,i} + X/V_i - (c_{Na^+,e} + c_{K^+,e} + c_{Cl^-,e}) \right) \right). \end{aligned} \quad (S32)$$

Following the approach from the mechano-osmotic flow analysis, we explore how an applied stress on the surface of one cell (associated with proliferation of surrounding cells) induces swelling of a neighboring cell. As per the single cell MEO analysis, the system of equations Eqns S23, S29-S32 can be solved either transiently or at steady state; both solutions are shown in Fig S6. Simulations reveal that the MEO model predicts the same trends as the mechano-osmotic model discussed in the main paper text (Fig 2): the volume of a loaded cell reduces with increasing applied stress and its neighbor increasingly swells due to an ion influx via gap junctions (Fig S6A). The mechanisms by which ions are driven into the neighbor differ slightly due to the consideration of electrical potentials. The applied stress generates an increase in the cytosolic hydrostatic pressure of the loaded cell, driving advective flow across GJs in accordance with Eqns S29-S31. As K^+ ions are the dominant species, such a flow increases the membrane potential of the connected cell. Due to the increased positive charge (Fig S6B), Cl^- ions are drawn into the neighboring cell from both the loaded cell and the external fluid (Fig S6E) to maintain electroneutrality. Solvent then enters the cell to reduce the osmotic pressure gradient, which is further lowered by the continuous influx of fluid from the advective GJ flow (Fig S3G). Further, the

loss of ions from the loaded cell impairs its ability to retain water relative to an isolated cell (Fig S5), again similar to the mechanisms identified from non-electrolyte flow in Fig 2.

Fig S6: Role of gap junctions (GJ) in cellular volume control in accordance with MEO model: In response to applied loading, predictions are shown for A) cell volume, B) membrane potential, C-E) concentrations of Na^+ , K^+ , and Cl^- , F) hydrostatic pressure and G) osmotic pressure at steady state in a loaded and connected unloaded cell. Markers show corresponding predictions at steady state from transient analysis.”

R3.3: At which timescale the assumed elastic constitutive relation of the cortex is true ? The cortex turnovers through actin polymerization/polymerization in a few minutes. At a long timescale relevant for growth, why isn't the cortical constitutive behaviour viscous ?

We have addressed a similar concern in response to Reviewer 2 (R2.2), and we reiterate some of our response here. If we view the cortex as a viscoelastic material, it may be considered to have an effective initial and long-term elastic modulus whereby although it will exhibit stress relaxation the stress will not entirely dissipate (actomyosin networks do not necessarily relax to a stress-free configuration⁴⁰). Also, as the timescales associated with volume equilibration (10s of minutes) are larger than cortical dissipation (10s of seconds), we may therefore neglect the influence of material viscosity and view the effective stiffness as a long-term (relaxed) modulus. Please note the following text in the revised manuscript:

“As per previous cortex models¹⁸⁻²⁰, we begin with the consideration of a general viscoelastic model such that the passive stress may be expressed by $\sigma_{p,i} = K(A_i/A_0 - 1)/2 + \eta(1/A_i)(dA_i/dt)$, where K is the effective cortical stiffness, η is the effective viscosity of the cell cortex, and A_0 is a reference surface area. The apparent cortex viscosity has been reported to lie in the range of $10^2 - 10^3 Pa \cdot s$ ^{21,22}, and under loading the cell radius has been reported to change by $\sim 10\%$ in several minutes²³. As such, the viscous stress $\eta(2/r)(dr/dt) \approx 0.1 - 1 Pa$ is much smaller than the elastic terms in the passive stress expression. Therefore, we consider the contribution of the viscous term to be negligible in this analysis with a view that K denotes the long-term cortical stiffness, and the total stress reduces to $\sigma_i = (K/2)(r_{c,i}^2/r_o^2 - 1) + \sigma_{a,i}$ for a spherical cell of radius $r_{c,i}$.”

Further we note that assuming the cortex as elastic, or even viscoelastic, is an idealization. Realistically the cytoskeleton actively remodels, with myosin motors actively generating force as dependent on stress-mediated signaling pathways. We have extensively considered such interactions and feedback in previous work^{24,25}, but within the current study these additional considerations would add a level of complexity that would distract from the key mechanisms of multicellular volume regulation, and we

therefore simply assume a constant active stress in the current framework. We intend to explore the evolution of active forces and signaling feedback with ion transport in future studies. Please note the following text the revised manuscript:

“An investigation more directed to the specific influence of cortical organization and myosin contractility could readily extend this expression to describe long-term remodeling in response to signaling and stress in accordance with our previous work^{24,25}.”

“Future advancements should also focus on detailing the interdependence between dynamic actomyosin contractility and cell osmolarity, building on our previous work to further understand how the two-way feedback between stress and signaling guides nuclear gene expression²⁶, dynamic force generation²⁷, and cancer invasion²⁸.”

R3.4: The authors recall the result of Ref. 26:

'MS channels are permeable due to tension in the cell membrane, permitting a constant loss of ions to the external media (Fig S1D). However, active ion pumping ensures there is a continuous ion influx (Fig S1F) to maintain the cell's osmotic pressure higher than that of the external media, allowing it to retain water'.

I think that one should be careful before including this complex model (many parameters and a non-linearity) in a larger framework for the following reasons. First, this model of cell volume regulation would need to be experimentally demonstrated. This has not been done in Ref. 26 (which is nonetheless useful from a theoretical standpoint) as in particular the cell filtration coefficient that they fit is several orders of magnitudes higher than the one reported in experiments, even considering their note in their table I. Second, the electro-static side of the problem has not been taken into account. While the number of macromolecules trapped in the cell is indeed small compared to the one of ions and does not contribute significantly to the osmotic pressure, their negative fixed charges which need to be neutralized by counter ions create an osmotic pressure that strongly affects the motion of ions (See the work of Y. Mori). This model is therefore different from the 'textbook' pump and leak model (Alberts et al., Molecular biology of the cell, see also the works of C. Peskin and Y. Mori) which minimally introduces two different ions (sodium/potassium) with a different membrane permeabilities. In this picture, the work of volume regulating pumps on the cell membrane is to exchange two ions not to simply import or export a single ion specie. Can the authors then better relate their model of ion pumping to the actual biological way volume regulating pumps (typically the sodium/potassium pump) work?

We thank the reviewer for this insightful comment. We have now reduced our filtration coefficient $L_{p,m}$ to lie with experimentally measured ranges. Please note the following additional text in the revised manuscript:

“Across a number of cell types the membrane solvent permeability rate P_s has been reported to lie in the range of $10^{-4} - 10^{-5} \text{ m/s}$ ²⁹. Jiang and Sun (2013)¹⁹ highlighted that this can be related to the membrane water permeability factor such that $L_{p,m} = P_s \bar{V}_w / RT$, from which we ascertain an upper value of $L_{p,m} = 7 \times 10^{-12}$ used here.”

Further, it should be noted that in the original study from Jiang and Sun (2013), the authors directly model the experiments of Stewart *et al* (2011)²³, although we agree with the reviewer that this was not immediately evident in their manuscript. Their model has been further validated in subsequent experimental/computational studies (Xie *et al.* (2018)⁴¹).

We have now addressed electro-osmotic considerations in the revised manuscript, as highlighted in response to the reviewer's previous comment (R3.2), with a detailed supplementary analysis provided in Section S8 as motivated by the now cited work of Mori (2011)³⁷ and others³⁶. Our analysis suggests that our reduced mechano-osmotic model captures the key trends predicted by the extended framework. Please note the following excerpt from the revised manuscript:

“Importantly, in our main analysis we limit ourselves to the consideration of a single ion species and the associated channels and pumps. However, the influence of multiple charged species, membrane potential and electroneutrality is explored in SI Section S8, with our analysis suggesting that the reduced single-species framework captures key trends predicted by our a full mechano-electro-osmotic model.”

Further, within this analysis we have provided further discussion of the biological mechanisms of volume-regulating pumps. Simulations suggest that our consideration of a single pump captures the key trends introduced by the extended analysis; Namely that an increase in pump activity is predicted to drive an increase in cell volume (Fig S5H-I). Please note the following additional text in the revised manuscript:

“In this formulation, p is the strength of the pump current with associated values representing the 3:2 stoichiometry of the $Na - K$ ATPase. While earlier we assumed that pumping actively transports a single ion species across the membrane, the biological mechanism of $Na - K$ pumping is more complex. From Glitsch and Tappe (1995)³⁵, the free energy during the pumping action is $\Delta G = \Delta G_a + 3 \left(-F\phi_i + RT \ln \left(\frac{c_{Na^+,e}}{c_{Na^+,i}} \right) \right) + 2 \left(F\phi_i + RT \ln \left(\frac{c_{K^+,e}}{c_{K^+,i}} \right) \right)$, where ΔG_a is an energy input associated with hydrolysis of ATP. Provided this reaction is energetically favorable ($\Delta G < 0$), the pump transfers three Na^+ ions against their concentration gradient to the cytosol, and two K^+ from the external media into the cytosol. However, the net result of this complex pathway, which downstream is predicted to cause an overall increase in cytosolic osmotic pressure and cell volume (Fig S5H-I), is similar to the behavior captured by only considering an active influx in Eqn 6.”

R3.5: The authors write

'recall that water flow is partly driven (by) hydrostatic pressure differences'.

What is the magnitude of these pressures compared to the osmotic pressure variation ? A classical argument raised in this context is that a variation of 5mM of osmolarity which can be regarded as a fluctuation given that 300mM is a typical cell medium osmolarity leads to an osmotic pressure that is several orders of magnitude higher than 50 Pa. It seems difficult to reconcile this with the fact that hydrostatic pressure differences in this range would directly lead to transmembrane solvent flow that are not negligible.

In our initial analysis the role of advective flow was not considered. With our extended model, as motivated by the reviewer's comments and our more detailed thermodynamic motivation, advective flow is now described. As a result of this extension, variations in external hydrostatic pressure now have a non-negligible influence on cell volume and transmembrane solvent flow. In connected cells, this is shown to have a similar influence as a variance in applied solid stress. We further note our model suggests that increased external hydrostatic pressure in unconnected round cells would not directly lead to an increased net transmembrane solvent flow, as the driving forces for transmembrane flow are balanced by the load that the same external pressure applies on the cell membrane, resisting influx and expansion (Fig S8D). However, the resultant increase in cytosolic hydrostatic pressure may downstream influence signaling pathways associated with cell growth and volume control. Please note the following additional content in the revised manuscript:

“The introduction of a local hydrostatic pressure perturbation ($\delta P^{ext} = 1 \text{ kPa}$) is shown in Fig S8. In the absence of gap junctions, our model predicts that (unlike solid stress) an additional fluid pressure does not cause a change in cell volume (Fig S8D). Although the increase in external hydrostatic pressure should drive water into the cell, swelling is opposed by the compressive load that the same interstitial pressure imposes on the cell membrane. However, the internal hydrostatic pressure of the loaded cell increases (by a magnitude equal to δP^{ext}). When GJs become permeable, because the hydrostatic pressure of the loaded cell is higher than that of its neighbor, advective flow drives solutes into the neighboring cell, lowering its membrane potential (Fig S8B). The loaded cell then shrinks due to water loss, though the electroneutrality condition and electrical potentials limit the volume reduction (Fig S8A). This mechanism is similar to that outlined for solid stress loading in Section S8, whereby the increasing ion concentration in the neighboring cell causes it to swell. Of note, the magnitude of the volume differences from local hydrostatic (δP_i^{ext}) and solid stress ($\sigma_{g,i}$) loading are equivalent (Fig S8C). Further, the combined analysis in this and the previous section highlight that an equivalent osmotic and hydrostatic pressure load do not have an identical influence on cellular volume. In terms of volume reduction in the locally loaded cell, an osmotic pressure induced by a 5 mM solute perturbation ($\sim 10 \text{ kPa}$) is found to be broadly equivalent to a hydrostatic pressure 10-fold lower ($\delta P_i^{ext} = 1 \text{ kPa}$). However, the impact on neighboring connected cells is markedly different. The same behavior may be explored with the mechano-osmotic model presented in the main paper text by similarly perturbing the external hydrostatic pressure by δP^{ext} in Eqns 3 and 4. Similarly, the extension can also be incorporated to the continuum-level model via inclusion of δP^{ext} in the membrane-specific term within Eqn 9. In this instance the hydrostatic perturbation $\delta P^{ext}(r)$ would need to be spatially defined in a similar context to $\sigma_g(r)$.

Fig S8: Response of connected cells to a local differences in hydrostatic pressure: Influence on A) cell volume and B) membrane potential; C) Predicted difference in cell volumes ($\Delta V = V_2 - V_1$) for local hydrostatic (δP_i^{ext}) and solid stress ($\sigma_{g,i}$) loading with open GJs; D) Influence of a hydrostatic pressure perturbation on cell volume when GJs are closed.”

The magnitudes of the osmotic and hydrostatic pressure differences across the membrane are predicted to be similar (Fig S1 and S2). We also have provided an additional analysis to explore the impact of an osmotic perturbation (5 mM – 13 kPa) in the context of electro-osmotic flow (Section S9), and our model suggests that rapidly introducing such a perturbation has a low impact on cell volume and hydrostatic pressure (on the order of 20 Pa) due to rapid cell adaption. Please note the following excerpts from the revised manuscript:

“As shown in Fig S7A, the introduction of a small osmotic shock (5 mM) has a marked, albeit low, influence on the volume of the shocked cell. The sudden change in the external Na^+ concentration causes the osmotic pressure difference to sharply decrease, by $\delta \Delta \Pi_i = RT(5 \text{ mM}) \approx 13 \text{ kPa}$ (Fig S7B).

However, this perturbation is rapidly balanced by solvent efflux from the cell and solute influx. Overall, it is interesting to note that hydrostatic pressure change that corresponds to such an osmotic pressure fluctuation is quite low (Fig S7C), predicted to be on the order of 20 Pa. Naturally, this magnitude depends on effective cell stiffness and resistance to volume change. The steady state volume of the connected neighboring cell is unaltered as the diffusive potential (introduced by the slight variance in the shocked cell's solute concentration) is not sufficient to overcome the balance of electrical potentials. The same behavior may be explored with the mechano-osmotic model presented in the main paper text by similarly perturbing the external osmotic pressure Π^{ext} in Eqns 4 and 6.

Fig S7: *Response of connected cells to a sudden local osmotic shock: Influence on A) cell volume, B) the hydrostatic pressure difference ΔP and C) the osmotic pressure difference $\Delta\Pi$.*

R3, Minor issues:

1) 'Cell proliferation is regulated by volume,...' not clear, specify which volume (cytoplasm, nucleus) and give references. There was a recent Nat.Phys. review on this topic.

The dependency of cell progression through the cell cycle on size was demonstrated by Varsano *et al.* (2017)⁴². However, the mechanisms associated with the size checkpoint (e.g. cytoplasmic/nuclear volume) are not evident. We have now modified the original statement from the abstract and included the suggested reference in the revised discussion. Please note the following modified text:

“Cell-cycle progression is coupled with cell size, but in 3D clusters it remains unclear how multiple cells interact to control their volume.”

“In this study, we identified that ion flow through gap junctions promotes peripheral cell swelling in loaded breast cancer model. Cell-progression through the cell-cycle is reportedly dependent on cell size, though the mechanisms are not yet clear^{42,43}.”

2) 'energy consuming ion transporters,..' ion pumps would be more clear

We have modified to 'pumps' throughout the revised manuscript.

3) 'ion channels on the membrane that permit passive exchange between the cytosol and extracellular fluid' is a bit misleading as the solvent does not travel through these channels but mostly through aquaporins.

Please note the following modified text in the revised manuscript:

“In single cells, volume control involves an interplay between ion channels on the membrane that permit passive exchange of solutes between the cytosol and extracellular fluid and active ion pumps that move solutes against a concentration gradient^{44,45}”

4) 'precise control of the cytosolic ion concentration can increase or decrease cell volume' you could quote <https://doi.org/10.1016/j.bpj.2016.05.011>

The study has now been cited in the revised manuscript. Please note the following:

“...precise control of the cytosolic ion concentration can increase or decrease cell volume⁴⁶.”

5) 'cells at the core were highly compressed' a bit misleading again. Their volume is smaller but you have not measured if they are exposed to a larger stress than in the periphery

We have now modified the text throughout the manuscript to reflect that the cells reduce in size. Please note the following example in the revised manuscript:

“We identified that peripheral cells became more swollen as the cluster grew, and cells at the core reduced in size”

6) 'via Van't Hoffs equation...' mention that this is a dilute (and actually questionable) assumption in the cell

We have modified the text accordingly to highlight the dilute assumption. Please note the following:

“Assuming dilute conditions, the cell's internal osmotic pressure relates to the number of ions N_i in the cytosol via Van't Hoffs equation $\Pi_i = N_i RT / V_{c,i}$ ”

7) Why is there a dependance of Delta Pi_c in Pi_{ext} dependance ? Would an osmotic shock affect Delta Pi_c ?

We have addressed a similar concern in response to a comment from Reviewer 2 (**R2.4**) and reiterate some of those points in relation to this comment. The mechanism was not clearly expressed in the initial text. Active ion pumps will transport ions to/from the cytosol as long as the process is energetically favorable ($\Delta G < 0$); It cannot require more energy to move an ion against the concentration gradient than is provided by ATP hydrolysis. If the osmotic pressure difference exceeds a critical value (determined by $\Delta G = 0$) then it becomes increasingly likely that the pump will change direction (reversal potential)³⁵. Please note the following text in the revised manuscript:

“These ion pumps require an energy input, such as from ATP hydrolysis, to overcome the energetic barrier associated with moving ions against the concentration gradient. Following Jiang and Sun (2013)¹⁹, the free energy change associated with pumping action can be expressed as $\Delta G = RT \log (c_i / c_{ext}) - \Delta G_a$, where ΔG_a is an energy input is associated with hydrolysis of ATP. The ion flux associated with active pumping can then be written as $\dot{n}_{p,i} = \gamma' \Delta G$, where γ' is a permeation constant. Maintaining our dilute assumption, ΔG can be linearized as $\Delta G = RT (\Pi_i - \Pi^{ext}) / \Pi^{ext} - \Delta G_a$. We can therefore identify a critical osmotic pressure difference $\Delta \Pi_c$, determined when $\Delta G = 0$, such that $\Delta \Pi_c = \Pi^{ext} \Delta G_a / RT$ (noting that when $\Delta G > 0$ active pumping is no longer energetically favorable and the pumping direction will reverse³⁵). Thus the ion flux generated by active pumping by ion transporters can be expressed as $\dot{n}_{p,i} = \gamma (\Delta \Pi_c - \Delta \Pi_i)$, where γ is a rate constant.”

Therefore, a significant alteration to the external ion concentration would be anticipated to affect pumping.

8) 'typically lying in the range of 100 – 250 Pa.' Give refs. This is highly dependent of the cell type.

Motivated by a later comment (11), we now approximate the range directly from simulated growth of a spheroid embedded in hydrogel. References are also provided to support that stress varies spatially. Please note the following modified text in the revised manuscript:

“To estimate the distribution of solid stress in our multicellular clusters, we simulate spheroid growth in a hydrogel system (SI Section S12); Simulations suggest that the solid stress varies from ~550 Pa at the core to ~200 Pa at the periphery, which we apply in our analysis (Fig 3B).”

“Dolega *et al.* (2017)⁴⁷ demonstrated that the pressure in a proliferative cell cluster under applied loading is spatially non-uniform by experimentally measuring the deformation of polyacrylamide beads embedded within the spheroid; the core stress was identified to be approximately equal to the applied load with a 2- to 3-fold reduction at the periphery”

9) In Fig 3D, when you compare the theoretical results with the experimental data, how many parameters are adjusted in the model?

Following the suggestions of the other reviewers (R1.2, R2.3), an expanded motivation for all parameters is now included in Section S6.

10) 'Interestingly, it has been shown that such stress is spatially non-uniform across the cancerous structure' which stress ? radial or hoop component ?

Dolega *et al.* (2017)⁴⁷ positioned polyacrylamide beads across spheroids, and determined the effective local pressure acting on the beads. As opposed to a radial or hoop component, it may be viewed as an effective uniform compressive stress acting on the bead surface. Please note the following text in the revised manuscript:

“Proliferation of cells within a growing cluster generates solid compressive stresses, additionally compounded by matrix stretch and cell confinement. Interestingly, it has been shown that such local compressive stresses are spatially non-uniform across the cancerous structure^{47,48}.”

11) I find it questionable to assume a fixed spatial dependence of $\sigma_g(r)$, which I assume is the radial component of the solid stress (but what is the influence of the hoop component which cannot be zero unless the stress field is uniform?). The solid stress, stemming from a certain growth law that needs to be specified, should be coupled via force balance in the spheroid to intra and extra-cellular water flows (work of Netti and Jain). These flows also transport ions and above all, feedback on the stress distribution. The mechanical description of the problem at the continuum level should be made more clear.

We thank the reviewer for highlighting this point. To clarify, $\sigma_g(r)$ is the compressive stress / pressure acting on the surface of a cell located at position r in the simulated spheroid. Please note the following clarification in the revised manuscript:

“We consider this compressive stress $\sigma_g(r)$ to act uniformly on the surface of a cell located at position r in the spheroid.”

As suggested by the reviewer we now motivate this stress distribution through the simulation of spheroid growth. Please note the following additional text in the revised manuscript:

“In our experimental system single cells were seeded in Matrigel/alginate hydrogels, which had a shear modulus of approximately 300 Pa. The isolated cells proliferated to achieve a small cluster by day 3

(Fig S10A) and continued to grow into a larger cluster on day 5. To characterize the stresses introduced by growth, we develop a finite element model of the proliferative cluster and simulate its deformation of the surrounding hydrogel. We adopt the multiplicative decomposition of the deformation gradient \mathbf{F} into an elastic tensor \mathbf{F}_e and a growth tensor \mathbf{F}_g as proposed by Rodriguez *et al.* (1994)⁴⁹, such that $\mathbf{F} = \mathbf{F}_e \mathbf{F}_g$. The growth tensor can be expressed by $\mathbf{F}_g = \lambda_g \mathbf{I}$, where λ_g is the growth stretch and \mathbf{I} is the second order identity tensor. Cluster growth from a single cell to a spheroid with a diameter of approximately 66.8 μm (Fig S10A) identifies that $\lambda_g \approx 4.4$, assumed to increase linearly from days 0-5. With this definition of the growth tensor, we can then determine the elastic component of the deformation gradient via $\mathbf{F}_e = \mathbf{F} \mathbf{F}_g^{-1}$. The mechanical behavior of the hydrogel and spheroid may then be described by a Neo-Hookean hyperelastic formulation, with a Cauchy stress given by:

$$\boldsymbol{\sigma} = \frac{G}{J_e} \left(\bar{\mathbf{B}}_e - \frac{1}{3} \text{tr}(\bar{\mathbf{C}}_e) \mathbf{I} \right) + \kappa (J_e - 1) \mathbf{I}, \quad (\text{S36})$$

where J_e is the determinant of the elastic component of the deformation gradient, $\bar{\mathbf{B}}_e = J_e^{-\frac{2}{3}} \mathbf{F}_e \mathbf{F}_e^T$ and $\bar{\mathbf{C}}_e = J_e^{\frac{2}{3}} \mathbf{F}_e^T \mathbf{F}_e$ are the left and right Cauchy-Green tensors, respectively, G is the material shear modulus, and κ is the material bulk modulus. Our hydrogels have a shear modulus $G_h = 300 \text{ Pa}$ and we assume a bulk modulus of $\kappa_h = 500 \text{ Pa}$ in line with previously reported values⁵⁰. We assume an effective whole-cell shear and bulk modulus of $G_h = 385 \text{ kPa}$ and $\kappa_h = 833 \text{ Pa}$, in accordance with our previous work²⁴. Simulations suggest that with increasing cluster growth, the surrounding hydrogel becomes increasingly stretched (Fig S10B), such that on day 5 a pressure $P \approx 550 \text{ Pa}$ is applied (Fig S10C). Recently, Dolega *et al.* (2017)⁴⁷ demonstrated that the pressure in a proliferative cell cluster under applied loading is spatially non-uniform by experimentally measuring the deformation of polyacrylamide beads embedded within the spheroid; the core stress was identified to be approximately equal to the applied load with a 2- to 3-fold reduction at the periphery, approximately following a linear distribution at intermediate locations. Our FE model predictions therefore suggest that the pressure acting on (uniformly) on the surface of individual cells in our day-5 system linearly varies from $\sigma_g^{max} = 550 \text{ Pa}$ at the spheroid core to $\sigma_g^{min} = 200 \text{ Pa}$ at the periphery (Fig S10D). This solid compressive stress enters the continuum framework via an expansion of Eqn 3 with $\sigma(r) = (\Delta P(r) - \sigma_g(r))/2h_i$. Following the continuum analysis from the main manuscript (Fig 3), we find that our model provides excellent agreement with our experimentally observed nuclear volumes at days 3 and 5 of growth (Fig S10E-F). The experimentally measured volumes were observed to be spatially uniform on day 3, indicating the applied solid stress is also uniform and can be motivated directly from growth predictions (Fig S10C).

Fig S10: *Prediction of spheroid surface pressure due to growth: A) Cross-section images of GFP-NLS-labelled MCF10A cells at day 3 and 5. Scale bar 50 μm ; B) Predicted matrix deformation from growth simulations using finite element analysis. Contours show max principal stretch in the hydrogel; C) Predicted evolution of spheroid surface pressure imposed by hydrogel during growth; D) Applied solid growth stress $\sigma_g(r)$ is highest at the core and spatially non-uniform; Predicted and experimental spatial cell and nuclear volumes under control conditions on E) day 3 and F) day 5 ($n > 3$)."*

However, a framework whereby the growth, force balance, and solute/solvent flow are fully coupled as suggested by this minor comment, is beyond the scope of the work. While such an analysis is of interest, it should consider how cell proliferation depends on size⁴² and feeds back into the growth formulation; this would require an in-depth investigation and development on novel formulations, worthy of an independent study. Please note the following text in the revised manuscript:

"Future implementations should consider the feedback between mitosis, cell size, solute/solvent flow, and force balance, building on previous models for spheroid growth^{51,52}".

We again sincerely thank the reviewer for their detailed and insightful comments, which have significantly improved the quality of our manuscript.

References

1. Sinyuk, M., Mulkearns-Hubert, E. E., Reizes, O. & Lathia, J. Cancer connectors: Connexins, gap junctions, and communication. *Frontiers in Oncology* **8**, (2018).
2. Debnath, J. & Brugge, J. S. Modelling glandular epithelial cancers in three-dimensional cultures. *Nature Reviews Cancer* **5**, 675–688 (2005).
3. Han, Y. L. *et al.* Cell swelling, softening and invasion in a three-dimensional breast cancer model. *Nat. Phys.* (2019). doi:10.1038/s41567-019-0680-8
4. Mathias, R. T., White, T. W. & Brink, P. R. Chapter 3 The Role of Gap Junction Channels in the Ciliary Body Secretory Epithelium. *Current Topics in Membranes* **62**, 71–96 (2008).
5. Nielsen, M. S. *et al.* Gap junctions. *Compr. Physiol.* **2**, 1981–2035 (2012).

6. Gao, J. *et al.* Lens intracellular hydrostatic pressure is generated by the circulation of sodium and modulated by gap junction coupling. *J. Gen. Physiol.* **137**, 507–520 (2011).
7. Stavarache, I. *et al.* Electrical behavior of multi-walled carbon nanotube network embedded in amorphous silicon nitride. *Nanoscale Res. Lett.* **6**, 87 (2011).
8. Paszek, M. J. *et al.* Tensional homeostasis and the malignant phenotype. *Cancer Cell* **8**, 241–254 (2005).
9. Chaudhuri, O. *et al.* Extracellular matrix stiffness and composition jointly regulate the induction of malignant phenotypes in mammary epithelium. *Nat. Mater.* **13**, 970–978 (2014).
10. Däster, S. *et al.* Induction of hypoxia and necrosis in multicellular tumor spheroids is associated with resistance to chemotherapy treatment. *Oncotarget* **8**, 1725–1736 (2017).
11. Onsager, L. Reciprocal relations in irreversible processes. I. *Phys. Rev.* **37**, 405–426 (1931).
12. Kedem, O. & Katchalsky, A. Thermodynamic analysis of the permeability of biological membranes to non-electrolytes. *BBA - Biochim. Biophys. Acta* **27**, 229–246 (1958).
13. Staverman, A. J. Non-equilibrium thermodynamics of membrane. *Trans. Faraday Soc.* **48**, 176–185 (1952).
14. Gonzalez, N. P. *et al.* Cell tension and mechanical regulation of cell volume. *Mol. Biol. Cell* **29**, 2591–2600 (2018).
15. Adar, R. M. & Safran, S. A. Active volume regulation in adhered cells. *Proc. Natl. Acad. Sci.* **117**, 201918203 (2020).
16. Jou Chan, C. *et al.* Hydraulic control of mammalian embryo size and cell fate. *Nature* doi:10.1038/s41586-019-1309-x
17. Dasgupta, S., Gupta, K., Zhang, Y., Viasnoff, V. & Prost, J. Physics of lumen growth. *Proc. Natl. Acad. Sci. U. S. A.* **115**, E4751–E4757 (2018).
18. Smeets, B., Cuvelier, M., Pešek, J. & Ramon, H. The Effect of Cortical Elasticity and Active Tension on Cell Adhesion Mechanics. *Biophys. J.* **116**, 930–937 (2019).
19. Jiang, H. & Sun, S. X. Cellular pressure and volume regulation and implications for cell mechanics. *Biophys. J.* **105**, 609–619 (2013).
20. Tinevez, J. Y. *et al.* Role of cortical tension in bleb growth. *Proc. Natl. Acad. Sci. U. S. A.* **106**, 18581–18586 (2009).
21. Evans, E. & Yeung, A. Apparent viscosity and cortical tension of blood granulocytes determined by micropipet aspiration. *Biophys. J.* **56**, 151–160 (1989).
22. Koay, E. J., Shieh, A. C. & Athanasiou, K. A. Creep indentation of single cells. *J. Biomech. Eng.* **125**, 334–341 (2003).
23. Stewart, M. P. *et al.* Hydrostatic pressure and the actomyosin cortex drive mitotic cell rounding. (2011). doi:10.1038/nature09642
24. Shenoy, V. B., Wang, H. & Wang, X. A chemo-mechanical free-energy-based approach to model durotaxis and extracellular stiffness-dependent contraction and polarization of cells. *Interface Focus* **6**, 20150067 (2016).
25. McEvoy, E., Shishvan, S. S., Deshpande, V. S. & McGarry, J. P. Thermodynamic Modeling of the Statistics of Cell Spreading on Ligand-Coated Elastic Substrates. *Biophys. J.* **115**, 2451–2460 (2018).
26. Alisafaei, F., Jokhun, D. S., Shivashankar, G. V & Shenoy, V. B. Regulation of nuclear

- architecture, mechanics, and nucleocytoplasmic shuttling of epigenetic factors by cell geometric constraints. *Proc. Natl. Acad. Sci. U. S. A.* **116**, 13200–13209 (2019).
27. McEvoy, E., Deshpande, V. S. & McGarry, P. Transient active force generation and stress fibre remodelling in cells under cyclic loading. *Biomech. Model. Mechanobiol.* **18**, 921–937 (2019).
 28. Ahmadzadeh, H. *et al.* Modeling the two-way feedback between contractility and matrix realignment reveals a nonlinear mode of cancer cell invasion. *Proc Natl Acad Sci U S A.* **2017**;114(9)E1617-E26. Epub **2017/02/16**, E1617–E1626 (2017).
 29. Marrink, S.-J. & Berendsen, H. J. C. *Simulation of Water Transport through a Lipid Membrane.* *J. Phys. Chem* **98**, (1994).
 30. Kumar, R., Saha, S. & Sinha, B. Cell spread area and traction forces determine myosin-II-based cortex thickness regulation. *Biochim. Biophys. Acta - Mol. Cell Res.* **1866**, 118516 (2019).
 31. Kuznetsova, T. G., Starodubtseva, M. N., Yegorenkov, N. I., Chizhik, S. A. & Zhdanov, R. I. Atomic force microscopy probing of cell elasticity. *Micron* **38**, 824–833 (2007).
 32. Alberts, B. *et al.* *Molecular biology of the cell.* (Garland Science, 2002).
 33. Grosell, M. Intestinal anion exchange in marine fish osmoregulation. *Journal of Experimental Biology* **209**, 2813–2827 (2006).
 34. Larsen, E. H., Møbjerg, N. & Nielsen, R. Application of the Na⁺ recirculation theory to ion coupled water transport in low- and high resistance osmoregulatory epithelia. *Comparative Biochemistry and Physiology - A Molecular and Integrative Physiology* **148**, 101–116 (2007).
 35. Glitsch, H. G. & Tappe, A. Change of Na⁺ pump current reversal potential in sheep cardiac Purkinje cells with varying free energy of ATP hydrolysis. *J. Physiol.* **484**, 605–616 (1995).
 36. Keener, J. P. & Sneyd, J. *Mathematical physiology.* (Springer, 1998).
 37. Mori, Y. Mathematical properties of pump-leak models of cell volume control and electrolyte balance. *J. Math. Biol.* **65**, 875–918 (2012).
 38. Patel, A., Demolombe, S. & Honoré, E. Piezo1 Ion Channels: An alternative to force. *Elife* **4**, (2015).
 39. Kay, A. R. How Cells Can Control Their Size by Pumping Ions. *Front. Cell Dev. Biol.* **5**, 41 (2017).
 40. Vigliotti, A., Ronan, W., Baaijens, F. P. T. & Deshpande, V. S. A thermodynamically motivated model for stress-fiber reorganization. *Biomech. Model. Mechanobiol.* (2015). doi:10.1007/s10237-015-0722-9
 41. Xie, K., Yang, Y. & Jiang, H. Controlling Cellular Volume via Mechanical and Physical Properties of Substrate. *Biophys. J.* **114**, 675–687 (2018).
 42. Varsano, G., Wang, Y. & Wu, M. Probing Mammalian Cell Size Homeostasis by Channel-Assisted Cell Reshaping. *Cell Rep.* **20**, 397–410 (2017).
 43. Cadart, C., Venkova, L., Recho, P., Lagomarsino, M. C. & Piel, M. The physics of cell-size regulation across timescales. *Nature Physics* **15**, 993–1004 (2019).
 44. Alberts, B. *et al.* Ion Channels and the Electrical Properties of Membranes. in *Molecular Biology of the Cell* (Garland Science, 2002).
 45. Okada, Y. Ion channels and transporters involved in cell volume regulation and sensor mechanisms. *Cell Biochemistry and Biophysics* **41**, 233–258 (2004).

46. Hui, T. H. *et al.* Regulating the Membrane Transport Activity and Death of Cells via Electroosmotic Manipulation. *Biophys. J.* **110**, 2769–2778 (2016).
47. Dolega, M. E. *et al.* Cell-like pressure sensors reveal increase of mechanical stress towards the core of multicellular spheroids under compression. *Nat. Commun.* **8**, (2017).
48. Nia, H. T. *et al.* *Quantifying solid stress and elastic energy from excised or in situ tumors.* (Nat Protoc 13, 2018).
49. Rodriguez, E. K., Hoger, A. & McCulloch, A. D. Stress-dependent finite growth in soft elastic tissues. *J. Biomech.* **27**, 455–467 (1994).
50. Steinwachs, J. *et al.* Three-dimensional force microscopy of cells in biopolymer networks. *Nat. Methods* **13**, 171–176 (2016).
51. Roose, T., Netti, P. A., Munn, L. L., Boucher, Y. & Jain, R. K. *Solid stress generated by spheroid growth estimated using a linear poroelasticity model.* (Microvasc Res 66, 2003).
52. Ambrosi, D., Pezzuto, S., Riccobelli, D., Stylianopoulos, T. & Ciarletta, P. *Solid Tumors Are Poroelastic Solids with a Chemo-mechanical Feedback on Growth.* *Journal of Elasticity* **129**, (J Elasticity 129, 2017).

REVIEWERS' COMMENTS

Reviewer #1 (Remarks to the Author):

Authors have adequately addressed all my comments and also important issues raised by the other reviewers. I feel that the manuscript, in its current format, is a great fit for publication in Nature communications.

Reviewer #2 (Remarks to the Author):

I am satisfied with the current version of the manuscript and I appreciate the efforts the authors made to clarify all the points raised by the 3 referees. It is definitely an interesting piece of modeling.

Reviewer #3 (Remarks to the Author):

I would like to thank the authors for extensively responding to my critical comments. The amount of work that has been performed to this end is sincerely impressive. I now find their modified model and supplementary analysis very complete and convincing. I do not fully agree with some specific points in their response but this is a matter of debate that does no longer belong to a review process. I am in favor of the manuscript publication.

We again thank all three reviewers for their constructive and supportive comments, and note there are no remaining remarks to be addressed.

Reviewer #1 (Remarks to the Author):

Authors have adequately addressed all my comments and also important issues raised by the other reviewers. I feel that the manuscript, in its current format, is a great fit for publication in Nature communications.

Reviewer #2 (Remarks to the Author):

I am satisfied with the current version of the manuscript and I appreciate the efforts the authors made to clarify all the points raised by the 3 referees. It is definitely an interesting piece of modeling.

Reviewer #3 (Remarks to the Author):

I would like to thank the authors for extensively responding to my critical comments. The amount of work that has been performed to this end is sincerely impressive. I now find their modified model and supplementary analysis very complete and convincing. I do not fully agree with some specific points in their response but this is a matter of debate that does no longer belong to a review process. I am in favor of the manuscript publication.